# Photochemical aging of atmospherically reactive organic compounds involving brown carbon at the air-aqueous interface

Siyang Li[1], Xiaotong Jiang[1], Marie Roveretto[3], Christian George[2,3], Ling Liu[1], Wei Jiang[1], Qingzhu Zhang[1], Wenxing Wang[1], Maofa Ge[4], Lin Du[1,*]

[1]Environment Research Institute, Shandong University, Binhai Road 72, Qingdao 266237, China
[2]School of Environmental Science and Engineering, Shandong University, Binhai Road 72, Qingdao 266237, China
[3]University of Lyon, Université Claude Bernard Lyon 1, CNRS, IRCELYON, F-69626 Villeurbanne, France
[4]State Key Laboratory for Structural Chemistry of Unstable and Stable Species, CAS Research/Education Center for Excellence in Molecular Sciences, Institute of Chemistry, Chinese Academy of Sciences, Beijing 100190, China

*Correspondence to*: Lin Du (lindu@sdu.edu.cn)

**Abstract.** Water-soluble brown carbon in the aqueous core of aerosol may play a role in the photochemical aging of organic film on the aerosol surface. To better understand the reactivity and photochemical aging processes of organic coating on the aqueous aerosol surface, we have simulated the photosensitized reaction of organic films made of several long chain fatty acids in a Langmuir trough in the presence or absence of irradiation. Several chemicals (imidazole-2-carboxaldehyde and humic

acid), $PM_{2.5}$ samples collected from the field and secondary organic aerosols samples generated from a simulation chamber were used as photosensitizers to be involved in the photochemistry of the organic films. Stearic acid, elaidic acid, oleic acid and two different phospholipids with the same carbon chain length and different degrees of saturation i.e., 1,2-distearoyl-sn-glycero-3-phosphocholine (DSPC) and 1,2-dioleoylsn-glycero-3-phosphocholine (DOPC) were chosen as the common organic film-forming species in this analysis. The double bond (*trans* and *cis*) in unsaturated organic compounds has an effect on the

surface area of the organic monolayer. The oleic acid (OA) monolayer possessing a *cis* double bond in an alkyl chain is more expanded than elaidic acid (EA) monolayers on artificial seawater that contain a photosensitizer. Monitoring the change in the relative area of DOPC monolayers has shown that DOPC does not react with photosensitizers under dark conditions. Instead, the photochemical reaction initiated by the excited photosensitizer and molecular oxygen can generate new unsaturated products in the DOPC monolayers, accompanied by an increase in the molecular area. The DSPC monolayers did not yield

any photochemical oxidized products under the same conditions. The spectra measured with polarization modulation-infrared reflection absorption spectroscopy (PM-IRRAS) were also consistent with the results of a surface pressure-area isotherm. Here, a reaction mechanism explaining these observations is presented and discussed. The results of $PM_{2.5}$ and SOA samples will contribute to our understanding of the processing of organic aerosol aging that alters the aerosol composition.

## 1 Introduction

In the marine environment, the degradation of bacteria and diatoms can produce lipids, such as phospholipids, triacylglycerides, and glycolipids, which are the lipid-containing cellular components of microorganisms (Jeffrey, 1966). Phospholipids can be transported from seawater into sea spray aerosol (SSA) directly (Collins et al., 2016). They also can be further transformed into fatty acids through heterotrophic breakdown (Li-Beisson et al., 2019; Guschina and Harwood, 2006). Long chain saturated fatty acids, such as palmitic acid and stearic acid, correspond to major constituents of the sea surface microlayer and are also detected in marine aerosols (Marty et al., 1979; Slowey et al., 1962; Wu et al., 2015; Hu et al., 2018; Kang et al., 2017). Unsaturated fatty acids with carbon chain lengths of 18-22 carbons also dominate the organic composition of seawater samples (Jeffrey, 1966; Osterroht, 1993). Oleic acid and elaidic acid were the most abundant unsaturated acids in seawater (Osterroht, 1993). The abundance of saturated fatty acids in $PM_{10}$ samples collected from the southeast coast of China was significantly higher than unsaturated fatty acids due to their enhanced photochemical stabilities and extensive atmospheric sources (Wu et al., 2015).

"Brown carbon" (BrC) refers to the type of light-absorbing organic aerosol material that efficiently absorbs radiation from visible to ultraviolet wavelengths in the atmosphere (Laskin et al., 2015; Andreae and Gelencser, 2006; Poschl, 2005). The fully dissolved organic fraction of BrC is referred to as light-absorbing water-soluble organic carbon (WSOC), while colloidal aggregates belong to water-insoluble organic carbon (WIOC). BrC can be generated from primary source, such as fossil fuel combustion and biomass burning. Multiple atmospheric reactions between gas and particle or aqueous phase also lead to secondary BrC production. The products from aqueous OH oxidation of phenolic compounds have characteristic BrC absorption spectra (Gelencser et al., 2003; Vione et al., 2014). The SOA generated by high-NOx photooxidation and ozonolysis of monoterpenes have absorption in the wavelength range of 355−780 nm (Nakayama et al., 2013; Flores et al., 2014). Aerosol flow tube experiments suggest that the exposure of SOA particles produced from biogenic alkenes to gaseous ammonia could increase the complex refractive index (Flores et al., 2014). The addition of amino acids or ammonium ions to limonene SOA was also shown to produce BrC, which strongly absorbed visible light and fluoresced at visible and ultraviolet wavelengths (Bones et al., 2010). Typical BrC is a mixture of a large number of strongly absorbing chromophores that have unique molecular structures and evolved from non-absorbing precursors (Nguyen et al., 2012; Lee et al., 2013).

Photosensitizers in the atmosphere can absorb and convert the energy of photons into chemical energy that can facilitate reactions at aerosol surfaces (George et al., 2015). Therefore, photosensitizers can contribute to organic aerosol aging and growth when generating a triplet excited state that can oxidize hydrocarbon upon absorbing light. These photosensitized reactions have been shown in the laboratory to induce to an abiotic source of volatile organic compounds (VOCs) and secondary organic aerosols (SOA) in the marine boundary layer (Rossignol et al., 2016; Tinel et al., 2016; Bernard et al., 2016). Traces of photosensitizing species in the aerosol phase, such as imidazoles, quinones and nitrophenols also contribute to SOA formation through their promoting effect with radical reactions (Li et al., 2016; Desyaterik et al., 2013; Pillar and Guzman, 2017).

Imidazole-2-carboxaldehyde (IC) is a component of BrC (Ackendorf et al., 2017; Arroyo et al., 2018; Rossignol et al., 2014), produced through the aqueous reaction of glyoxal or methylglyoxal with ammonium sulphate (Aiona et al., 2017). Recent works suggest that IC can act as an efficient photosensitizer in autophotocatalytic aerosol growth (Aregahegn et al., 2013; Rossignol et al., 2014; Palacios et al., 2016). For example, aerosol seeds containing IC can induce the reactive uptake of VOCs when exposed to light, leading to SOA formation (Aregahegn et al., 2013). In addition to small-molecular-weight species like IC, water-soluble organic materials with higher molecular weight like humic-like substances (HULIS) can also absorb light. HULIS have the similar properties to macromolecular humic substances, such as their amphiphilic and polyacidic nature, aromaticity, surface active properties and light absorption ability (Gelencser et al., 2002; Graber and Rudich, 2006; Sannigrahi et al., 2006; Krivacsy et al., 2008). HULIS correspond to 10−35% of fine organic materials in atmospheric aerosols and account for up to 72% of WSOC in some ambient aerosol samples (Emmenegger et al., 2007; Feczko et al., 2007). Humic substances (HS) consist of three operationally defined components: humic acids (HA), fulvic acids, and humins. These HS represent a fraction of the molecularly uncharacterized component of dissolved organic matter in the ocean (Zhu et al., 2017b). HA in the ocean are widely believed to derive primarily from the products of marine phytoplankton degradation and less so from terrestrial sources (McCarthy et al., 1996). For primary marine aerosols, WSOC containing HULIS components was suggested to originate from bubble-bursting at the surface of seawater, which transfers organic matter into marine aerosol particles (Cavalli et al., 2004; Yu et al., 2004). HULIS exist primarily in the droplet mode with aerodynamic diameter in the range of 0.7-0.8 μm (Wang and Yu, 2017).

Atmospheric aerosol particles are a mixture of a large number of inorganic and organic compounds. The different physico-chemical properties of aerosol components result in inorganic-salt aerosols being coated with a film of surface-active organic materials (Raymond and Pandis, 2003). The structure of the organic coated aqueous aerosol can be simplified as an aqueous core-shell model (Song et al., 2013; Tervahattu et al., 2005; Ellison et al., 1999). The presence of organic film on the aerosol surface can reduce the evaporation rate of water from droplets or particles (Nguyen et al., 2017; Davies et al., 2013). The transport and uptake of chemicals between the gas and liquid phase is affected by organic coating (Donaldson and Valsaraj, 2010). The packing order and stability of films on the surface also facilitate further discussion of the uptake efficiency of marine aerosols toward atmospheric trace gases such as $N_2O_5$ and $HNO_3$ (Bertram et al., 2018). The organic film can reduce the scavenging by larger cloud droplets and affect the lifetime of aerosols (Toossi and Novakov, 1985; Gill et al., 1983). In addition, the optical properties of the droplet can be altered by organic coating (Donaldson and Vaida, 2006).

Atmospheric aerosols may contain light-absorbing BrC in the aqueous core. BrC, which is excited by visible and ultraviolet radiation in the sunlight, can act as photosensitizer to facilitate free radical reactions and contribute to photochemical aging of organic aerosol (Smith et al., 2014; Yu et al., 2014). The photosensitized uptake of limonene was also observed for several SOA materials containing a complex mixture of photosensitive organic compounds collected from the chamber (Malecha and Nizkorodov, 2017). When aerosols containing photosensitizers such as IC and HA were exposed to isoprene, limonene, α-pinene, β-pinene and toluene in the presence of UV light, there was significant SOA formation (Palacios et al., 2016). The gaseous products from interfacial photochemistry of organic film can act as precursors for SOA, while highly

oxidized products are likely to be more water-soluble (Bruggemann et al., 2017; Alpert et al., 2017). The interfacial photochemistry of a palmitic acid monolayer involving HA as a photosensitizer has been recently investigated (Shrestha et al., 2018). Therefore, the photochemistry of such organic films is attracting increasing attention, as the photosensitized reaction initiated by atmospheric samples containing BrC has not been investigated and the associated mechanism is not understood.

5    Here, we investigated the interfacial photochemical properties of WSOC from three different sources: i.) IC and HA as identified proxies; ii.) SOA samples collected from a simulation chamber (regarded as a half-real atmospheric sample); and iii.) authentic PM$_{2.5}$ samples. To the best of our knowledge, the photosensitized reaction of organic Langmuir films involving chamber and atmospheric samples has not been previously investigated. Fatty acids and phospholipids such as 1,2-distearoyl-sn-glycero-3-phosphocholine (DSPC), 1,2-dioleoylsn-glycero-3-phosphocholine (DOPC), stearic acid (SA), elaidic acid (EA) and oleic acid (OA) were selected as proxies of low volatile organic compounds of organic aerosols. Langmuir monolayers at the air-aqueous interface were employed to mimic the organic films that coat the aqueous aerosols (Sobanska et al., 2015; Ruehl and Wilson, 2014; King et al., 2009; Sebastiani et al., 2018; Adams et al., 2016). The stability behaviour of these organic films under irradiation and the impact of photosensitizers on organic aerosol aging are presented and discussed.

## 2 Experimental section

### 2.1 Materials and solutions

The zwitterionic phospholipids (DSPC and DOPC) were acquired from Avanti Polar Lipids Inc. The fatty acids (SA, EA and OA) were purchased from Sigma-Aldrich. All chemicals used in the experiment did not require further purification. Ultrapure water (18.2 MΩ•cm) provided by a Milli-Q system (Millipore, France) was used to prepare the artificial seawater (ASW), which is a solution of multiple salts: 426 mM NaCl (≥99%, Acros), 55.5 mM MgCl$_2$ (99.9%, Alfa Aesar), 29.4 mM Na$_2$SO$_4$ (99%, Alfa Aesar), 10.8 mM CaCl$_2$ (99%, Adamas), 9.45 mM KCl (99%, Alfa Aesar), 2.43 mM NaHCO$_3$ (≥99.7%, Alfa Aesar), 0.86 mM KBr (≥99%, Acros), 0.44 mM H$_3$BO$_3$ (99.5%, Innochem), 0.074 mM NaF (≥99%, Acros), and 0.094 mM SrCl$_2$•6H$_2$O (≥99%, Alfa Aesar) (Brzozowska et al., 2012; Brzozowska et al., 2013). The effect of artificial sea water on stabilizing organic monolayer has been confirmed in previous studies (Li et al., 2018; Li et al., 2019).The photosensitizers imidazole-2-carboxaldehyde (IC) (97%, Alfa Aesar) or HA (> 90% fulvic acid, Aladdin) were added into the ASW with concentrations of 2.5 mM and 30 mg/L, respectively. Five different subphases were used for our experiments: pure ASW, ASW with 2.5 mM IC, ASW with 30 mg/L HA, ASW with PM$_{2.5}$ sample, and ASW with SOA sample. The molecular structures of DOPC, DSPC, SA, EA and OA are provided in **Fig. 1(a)**.

### 2.2 Atmospheric PM$_{2.5}$ sample collection and preparation

Atmospheric fine particulate matter samples were collected at the atmospheric observation station (36°40' N, 117°03' E) located in the central campus of Shandong University in January, 2016. A medium-volume PM$_{2.5}$ sampler (TH-150 A, Wuhan Tianhong, China) was used to collect PM$_{2.5}$ samples on Isopore™ membrane filters (90 mm diameter, 0.4 μm pore

size; HTTP, Merck Millipore, Germany). The flow rate of sampling was 100 L/min. The filter samples were sealed and stored at −20 °C before the experiment. All the filters were dissolved into 40 mL ultrapure water with ultrasonic agitation. Sonication was performed in an ultrasonic bath with a frequency of 40 kHz and the power of 80 W. The sonication time was 15 min. Subsequently, the suspension was centrifuged at 1780 g for 40 min. The supernatant, which contains the water-soluble fraction including water-soluble organic compounds (WSOC) and inorganic ions, was re-collected by freeze-drying. The insoluble fraction separated from soluble fraction was also freeze-dried. The mass ratio of insoluble to soluble fraction is 0.91:1. Then, 3.3 mg of freeze-dried soluble sample was dissolved in 1000 mL artificial seawater. The concentration of $PM_{2.5}$ sample in the artificial seawater is 3.3 mg/L.

### 2.3 Chamber sample preparation

The laboratory SOA samples were generated from a 1 $m^3$ FEP-Teflon film chamber at room temperature (298 ± 2 K) and atmospheric pressure. The details of the chamber including the experimental setup and the schematic of the smog chamber facility have been previously described (Liu et al., 2019; Wang et al., 2018; Zhu et al., 2017a). To maximize and homogenize the intensity of radiation, stainless steel mirror surfaces were used as the interior wall of the chamber enclosure. The chamber was surrounded by a total of twelve black lamps (Philips TUV 36 W) with a peak intensity at 365 nm to simulate actinic UV irradiation. Purified air produced by a zero air gas generator (111-D3N, Thermo Scientific, USA) was used as background. Prior to each experiment, the chamber was cleaned by zero air for more than 4 h. Liquid droplet ammonium sulphate seed particles were injected into the chamber by nebulizing a 0.6 M ammonium sulphate (($NH_4$)$_2SO_4$) aqueous solution with an aerosol generator (Model 3076, TSI, USA). A customized diffusion dryer was added after aerosol generator to make sure that the ($NH_4$)$_2SO_4$ aerosols were in solid phase in chamber. Relative humidity was about 20 % throughout the experiments, which is lower than the crystallization RH (35%) of ($NH_4$)$_2SO_4$ (Ng et al., 2007). The seed particle was kept at solid phase. The total gas volume in the chamber was recorded with mass flow meters (D80-8C/ZM, Beijing Sevenstar, China). Limonene (99%, tci) was injected into the chamber by a micro syringe and was evaporated into a stream of purified air. Then, an aqueous $H_2O_2$ solution (30 wt %) was injected to the chamber and served as the OH precursor in these experiments. The concentration of $H_2O_2$ was estimated to be 4324 ppb. NO was introduced into the chamber by a gas-tight syringe. Typically, 684 ppb limonene and 5×10$^4$ cm$^{-3}$ ($NH_4$)$_2SO_4$ seed aerosols were employed. The concentration of limonene was determined by gas chromatograph equipped with flame ionization detector (GC-FID) (Agilent Technologies, GC-FID 7890B). The SOA formation of limonene photooxidation experiments was performed under high-$NO_x$ condition (Sarrafzadeh et al., 2016). The initial concentration of $NO_x$ and NO detected by NO-$NO_2$-$NO_x$ analyzer (Model 42C, Thermo Electron Corporation, USA) were 206 ppb and 164 ppb, respectively. The reaction was allowed to proceed for 4 hours before SOA particles were collected. At that time, the concentration of limonene could not be detected. The initial concentrations of reactants in the chamber were also listed in **Table S1** in the supplement. The SOA particles were collected on pieces of aluminium foil with 25 mm diameter by using a 13-stage Dekati low-pressure impactor (DLPI) (DLPI+, Dekati Ltd, Finland). The particle sizes that were collected by DLPI at a sampling flow rate of 10 L min$^{-1}$ ranged from 16 nm to 10 μm. Then, all the aluminium foil substrates were combined for

extraction. The SOA samples collected on the aluminium foil pieces were dissolved in ultrapure water by sonicating for 1 min in an ultrasonic bath. The extract water solution was concentrated by rotary evaporation. The residue was dried under high purity nitrogen stream. Then, the SOA sample was transferred to the artificial seawater with the concentration of 0.66 mg/L.

## 2.4 Calculation of Mass Absorption Coefficient

The UV–vis absorption spectra of four photosensitizers were measured using a UV-vis spectrophotometer (P9, Shanghai Mapada, China). Spectra were collected using quartz cuvettes with internal path length of 1.0 cm. Aqueous solutions of 0.006 g/L IC, 0.006 g/L humic acid, 0.01 g/L PM$_{2.5}$ sample and 0.01 g/L SOA sample were used.

The IC aqueous solution displays a major absorption peak at 288 nm which is in an agreement with previous studies (Tinel et al., 2014; Berke et al., 2019). The maximum of absorption of SOA sample was at 286 nm, as can be seen from the
absorption spectrum shown in **Fig. S1** in the supplement. Similar to the previous studies, HA and PM$_{2.5}$ samples' absorptions have the main feature around 200 nm with a clearly visible hump between 250 and 300 nm (Kristensen et al., 2015). We calculate the mass-absorption coefficients (MAC) (cm$^2$/g) of four samples by the following equation:

$$MAC(\lambda) = \frac{A(\lambda) \times \ln 10}{b \times C_m},$$ (1)

where A is the absorption, b is the length of the cuvette (1 cm), and C$_m$ is the concentration of the dissolved reaction products
(g/mL).

In order to compare the light-absorbing properties of four samples, we estimated the light-absorbing properties of aerosols by the average MAC (<MAC>) over the wavelengths range from $\lambda_1 = 200$ nm to $\lambda_n = 700$ nm) (Jiang et al., 2019).

$$\langle MAC \rangle = \frac{\sum_{i=1}^{n} MAC(\lambda_i)}{n},$$ (2)

HA sample with <MAC>$_{200-700\ nm}$ of 64460 cm$^2$/g is also more absorbing than IC sample (<MAC>$_{200-700\ nm}$ = 41267 cm$^2$/g),
SOA sample (<MAC>$_{200-700\ nm}$ = 32867 cm$^2$/g) and PM$_{2.5}$ sample (<MAC>$_{200-700\ nm}$ = 16048 cm$^2$/g).

## 2.5 Langmuir and irradiation experiments

Langmuir and irradiation experiments were performed in a Langmuir trough (Riegler & Kirstein GmbH, Germany). The phospholipid and fatty acid monolayers were prepared by dropping 1 mM chloroform stock solution onto the artificial seawater containing the photosensitizer. After spreading the surfactant solution, ten to fifteen minutes were allowed for the
evaporation of solvent before compression and subsequent measurements. Both the surface pressure and molecular area of the Langmuir monolayers were measured with the Wilhelmy plate method during the barrier compression at a constant barrier speed of 3 mm/min. A Wilhelmy plate is a rectangular plate made of filter paper with a few centimeters in length and height. The plate is attached to a force sensor on the one side of barrier. In the measurement procedure, the Wilhelmy plate is first

dipped into the liquid and then pulled back to the position of first contact to measure the surface pressure (Rame, 1997; Hyvärinen et al., 2006; Aumann et al., 2010). The surface pressure was obtained based on the following equation:

$$\pi = \gamma_0 - \gamma, \tag{3}$$

where $\gamma_0$ is equal to the surface tension of the solution and $\gamma$ is the surface tension due to the monolayer. A Plexiglass cover was used to guarantee that the experimental condition was relatively sealed. All the surface pressure-area measurements were repeated at least three times to ensure consistent results (see for example **Fig. S2**). The change of surface pressure at the same molecular area was measured within ±2 mN/m. Standard deviation of the molecular area was ±1 Å$^2$/molecule. For area relaxation measurements, the changes of relative molecular area ($A/A_0$) with time were also recorded, where $A$ is the measured molecular area and $A_0$ is the initial area for DOPC monolayer when the compression to 25 mN/m was reached. The irradiation experiments were also conducted at the surface pressure equilibrium of 25 mN/m. The trough was illuminated by three UV lights (Philips TUV TL-Mini 8 W) with peak intensity at 365 nm. The lamps were located above the Langmuir trough at a distance of 20 mm. Pure water is relatively transparent to the near-infrared and near ultraviolet wavelengths of light. Dissolved salts in seawater have no light absorption under visible light but slight absorption under ultraviolet light (Carpenter and Nightingale, 2015). The irradiation and non-irradiation experiments were conducted with or without the photosensitizers.

**2.6 Polarization modulation-infrared reflection absorption spectroscopy measurements**

Polarization modulation-infrared reflection absorption spectroscopy (PM-IRRAS) (Bruker, Germany) experiments were conducted to detect the change of organic films at the air−aqueous interface in both dark and irradiation conditions. IRRAS spectra of organic film are generally presented as plots of reflectance-absorbance against wavenumber. Reflectance-absorbance is defined as $-\log_{10}(R/R_o)$ where R is the reflectivity of the film-covered surface and $R_o$ is the reflectivity of the aqueous subphase. The incident beam is directed onto the aqueous solution surface in the Langmuir trough at a 40° angle based on our previous study (Li et al., 2017b). Then, the reflected beam is measured by the MCT detector. The polarization modulation can modulate incident light into p polarization (perpendicular to the reflection plane) and s polarization (parallel to the reflection plane). The difference between the s-polarized and p-polarized spectra can provide information on the organic films at the air−aqueous interface. The spectra were recorded at both polarizations simultaneously due to the high scanning frequency of 42 kHz, thus eliminating the effects of the water vapour and carbon dioxide. When the surface pressure of the organic monolayer reached the set value of 25 mN m$^{-1}$, the IRRAS spectra were recorded in the range of 400-4000 cm$^{-1}$. Each spectrum averaged 2000 scans and had a resolution of 8 cm$^{-1}$. The variability in the IRRAS peaks over the three trials was smaller than 2 cm$^{-1}$, as shown in **Fig. S3**. In the irradiation experiment, first the black lamp was turned on at the surface pressure of 25 mN m$^{-1}$, starting irradiation. IRRAS spectra were measured when 1.5 h irradiation was finished. The experiments in the absence of irradiation were performed to guarantee that the shifts of band position are induced by photochemical oxidation and not related to spectra variability. **Fig. 1(b)** shows a schematic of the IRRAS and irradiation setups.

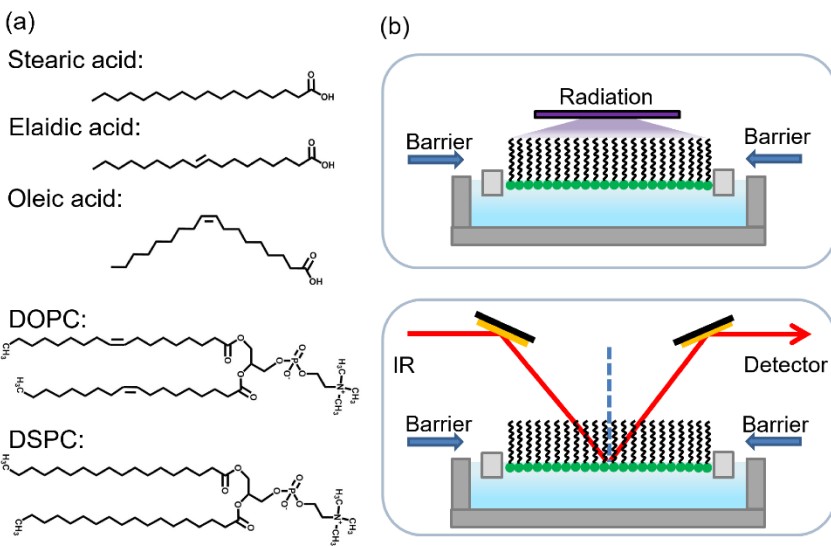

**Figure 1: (a) Molecular structures of stearic acid, elaidic acid, oleic acid, DOPC, and DSPC. (b) Schematic representation of IRRAS and irradiation setup.**

## 3 Results and discussion

### 3.1 Packing and phase behaviour of lipid monolayers

The monolayers usually composed of amphiphilic molecules with a hydrophilic head and a hydrophobic tail are assembled vertically at the air–water interface. Surface pressure–area ($\pi$−A) isotherms show phase transition behaviours of organic films. Owing to the amphiphilic characteristics of phospholipids, the head groups of DOPC and DSPC molecules prefer to be in the solution while their tails stretch into the air. The $\pi$−A isotherms recorded for DOPC and DSPC monolayers on artificial seawater with and without photosensitizers are shown in **Fig. 2(a)** and **(b)**, respectively. When the surface area of the DOPC monolayer was larger than 125 Å$^2$/molecule, the distance between the DOPC molecules was quite large and the intermolecular force was quite weak. The surface pressure of a DOPC monolayer on artificial seawater started to increase from 125 Å$^2$/molecule, where the DOPC monolayer surface state underwent a transition from the gas to the liquid-condensed phase. After the first phase transition, there is a proportional increase in surface pressure with decreasing area. This caused condensation and ordering at the interface, increasing the surface pressure of the organic monolayer. This trend continued up to a point where the DOPC molecules were packed closely and have very little space to move. Finally, the DOPC monolayer collapsed at 46 mN/m (Pereira et al., 2018). Applying an increasing pressure at the collapse pressure caused the monolayer to become unstable and destroy the monolayer. The packing and phase behaviours of DOPC monolayer at different pressures were also shown in the inserts in **Fig. 2(a)**.

The $\pi$−A isotherms of the DOPC monolayers were shifted to larger molecular areas upon the introduction of the photosensitizing molecules into the bulk artificial seawater. Due to their high-water solubility, pure IC and HA molecules

cannot form Langmuir monolayers. The collapse pressure of DOPC monolayer on pure artificial seawater decreased from 46 to 28 mN/m with the addition of 2.5 mM IC. The collapse pressure for the DOPC monolayer on the artificial seawater containing 30 mg/L HA was even lower. The DOPC monolayers on the artificial seawater containing water-soluble organic species extracted from PM$_{2.5}$ and SOA samples also shifted to larger areas. The collapse pressure for the DOPC monolayer on

the artificial seawater with PM$_{2.5}$ sample was 41 mN/m. The π–A isotherms of the DOPC monolayer on artificial seawater mixed with SOA sample and IC overlaps in the liquid-condensed phase. Finally, the curve the DOPC monolayer on artificial seawater containing SOA sample showed a collapse pressure similar to that on pure artificial seawater. In contrast to IC and HA, the collapse pressures of DOPC monolayer were higher in the PM$_{2.5}$ and SOA samples. The inorganic ions from the PM$_{2.5}$ and SOA samples like metal ions may contribute to the organization of organic monolayer (Adams et al., 2016; Adams et al.,

2017; Denton et al., 2019).

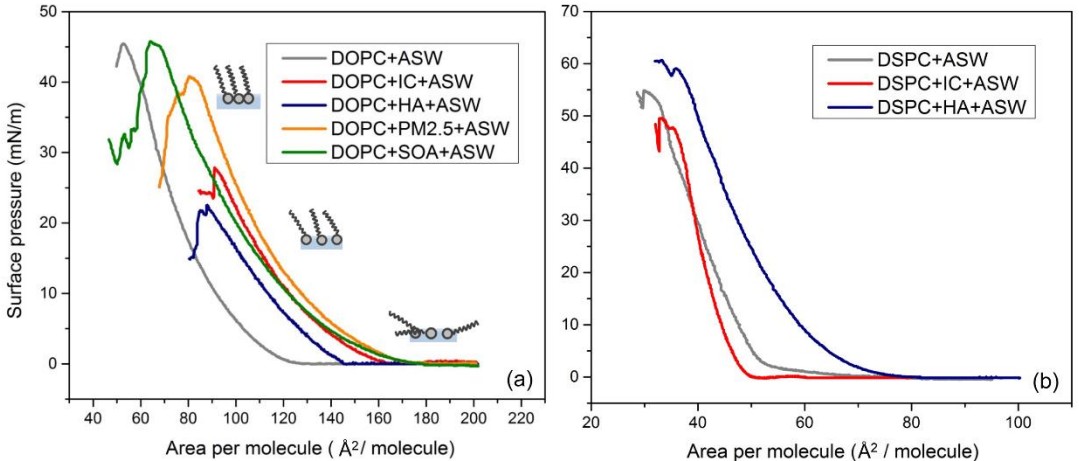

**Figure 2: Surface pressure–area (π–A) isotherms of (a) DOPC and (b) DSPC monolayers on artificial seawater (ASW) containing IC, HA, PM$_{2.5}$ and SOA samples. The insets show the phase behaviours of DOPC monolayer at different pressures.**

In line with previous studies, DSPC molecules pack closely and form numerous van der Waals bonds to hold the DSPC monolayer together, with a collapse pressure was 55 mN/m (Chou et al., 2002; Yin et al., 2016). The appearance of the secondary increase in surface pressure after DSPC monolayer collapsed, indicated that DSPC monolayer had a tendency to form multiple layer. The lift-off area is the area when the surface pressure of organic monolayer starts to increase. Although the structures of DSPC and DOPC are quite similar, the lift-off area for the DSPC monolayer on artificial seawater was smaller

than that for the DOPC monolayer in **Fig. 2(b)**. The difference of lift-off areas can be attributed to the unsaturated aliphatic chains in the DOPC molecule, which induce a less ordered structure relative to DSPC with saturated chains. The less tight packing of the hydrocarbon tail in DOPC could create more space for the incorporation of photosensitizer molecules, which is in good agreement to the earlier report (Aoki et al., 2016). The introduction of HA into the subphase also had a profound effect

on the shape of π−A isotherms for DSPC monolayers. At any fixed surface pressure, the molecular area for the DSPC monolayer on artificial seawater with HA was much larger than that for just the DSPC monolayer.

As shown in **Fig. 3**, the lift-off area for the SA monolayer is 40 Å²/molecule, which is substantially smaller than the lift-off areas for the EA and OA monolayers. Completely saturated stearic acid has a straight chain which can pack tightly into an untilted-condensed (UC) phase. The surface pressure of EA and OA monolayers began increasing at the same molecular area of 82 Å²/molecule. The unsaturation in the alkyl chains of EA and OA is responsible for the differences in the lift-off areas. Compared to a fully saturated SA with the same number of carbon atoms, elaidic acid and oleic acid molecules possessing a double bond cannot be fully condensed into monolayer. However, the EA monolayer can reach a maximum surface pressure of 34 mN/m, which is higher than that of OA (below 20 mN/m). Elaidic acid has a double bond at the number 9 carbon. The double bond of alkane chain in elaidic acid is *trans*, therefore elaidic acid molecule remains straight and can still pack well in a monolayer (Kanicky and Shah, 2002). Oleic acid is the *cis*-isomer of elaidic acid. The kink and bend of the alkyl chain at the position of the *cis* double bond induced steric hindrance during the compression of oleic acid molecules. The oleic acid monolayer was less able to pack as tightly as elaidic acid and form a tilted-condensed (TC) phase.

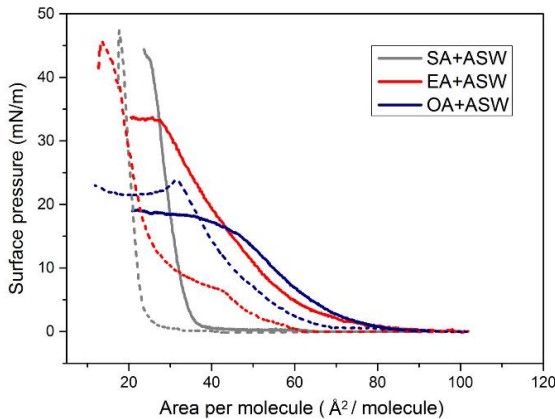

**Figure 3: Surface pressure–area (π–A) isotherms of SA, EA and OA monolayers on ASW (solid line) and ASW containing IC (short dashed line) solutions.**

In the presence of IC in the artificial seawater, the fatty acid monolayers shifted to smaller areas and had higher collapse pressures relative to pure artificial seawater. The lift-off area value of 60 Å²/molecule for EA monolayer on artificial seawater containing IC is larger compared to saturated SA monolayer on the same subphase (28 Å²/molecule). The EA monolayers on the artificial seawater containing IC transitioned from TC to UC phase at the surface pressure of about 7−8 mN/m during the monolayer formation. Additionally, the EA monolayer exhibits the TC and UC phase transition below 17 °C (Iimura et al., 2001). The EA monolayer on the artificial seawater containing IC collapsed at the surface pressure of 46 mN/m. The collapse of the OA monolayer on the artificial seawater containing IC occurred in the TC state at a surface pressure of 24

mN/m. The behaviour in the UC phase of the EA monolayer is similar to that of the SA monolayer, reflecting that the photosensitizer molecules may be squeezed out of the tail of EA at high surface pressure. There is a remarkable obstructive effect on the ordering of organic film for the film-forming molecule that contain a *cis*-double bond. The *cis*-double bond in OA disturbs close packing of the molecules and weakens the chain-chain attractive interaction (Iimura et al., 2001). Therefore,
OA has a greater tendency to form expanded monolayers.

## 3.2 Stability of lipid monolayers under irradiation

In order to get a better understanding of stability behaviour and relaxation mechanism for lipid monolayers at the air–water interface, we measured area relaxation curves at a fixed surface pressure of 25 mN m$^{-1}$ for both dark and irradiated experiments. The change in relative area ($A/A_0$) for the lipid monolayer was analysed as a function of time. $A_0$ represents the
initial area for DOPC monolayer at the surface pressure of 25 mN/m. The relative area curves of irradiated and non-irradiated DOPC monolayers are shown in **Fig. 4(a-e)**. The curve for the DOPC monolayer on pure artificial seawater exposed to the illumination was similar to that without irradiation. However, considerable changes of molecular areas for the DOPC monolayer were observed when the irradiation started in presence of a photosensitizer. The relative area of the DOPC monolayer on pure artificial seawater was reduced to 30% after 90 min irradiation. It is evident that the presence of
photosensitizing molecules in the subphases decreased the loss of the molecular area for the DOPC monolayer. In the case of artificial seawater containing IC and HA (**Fig. 4(b-c)**), the molecular area of the DOPC monolayer after 90 minutes of irradiation was reduced to 89% and 74%, respectively. The loss of molecular area for the DOPC monolayer on the artificial seawater containing the SOA sample was similar to that in artificial seawater containing HA. In the presence of artificial seawater mixed with the PM$_{2.5}$ samples, there was only a 5% loss of the molecular area of the DOPC monolayer after 90
minutes of irradiation. The results of these curves indicated that there was no photosensitized reaction with only artificial seawater solutions. The decreasing trend of the molecular area for the irradiated DOPC monolayer on artificial seawater was due to the loss of organic molecules in the subphase. The rapid loss of molecular area on pure artificial seawater was related to spontaneous lipid degradation by oxidation, which was induced by reactive oxygen species in the air (Liljeblad et al., 2010).

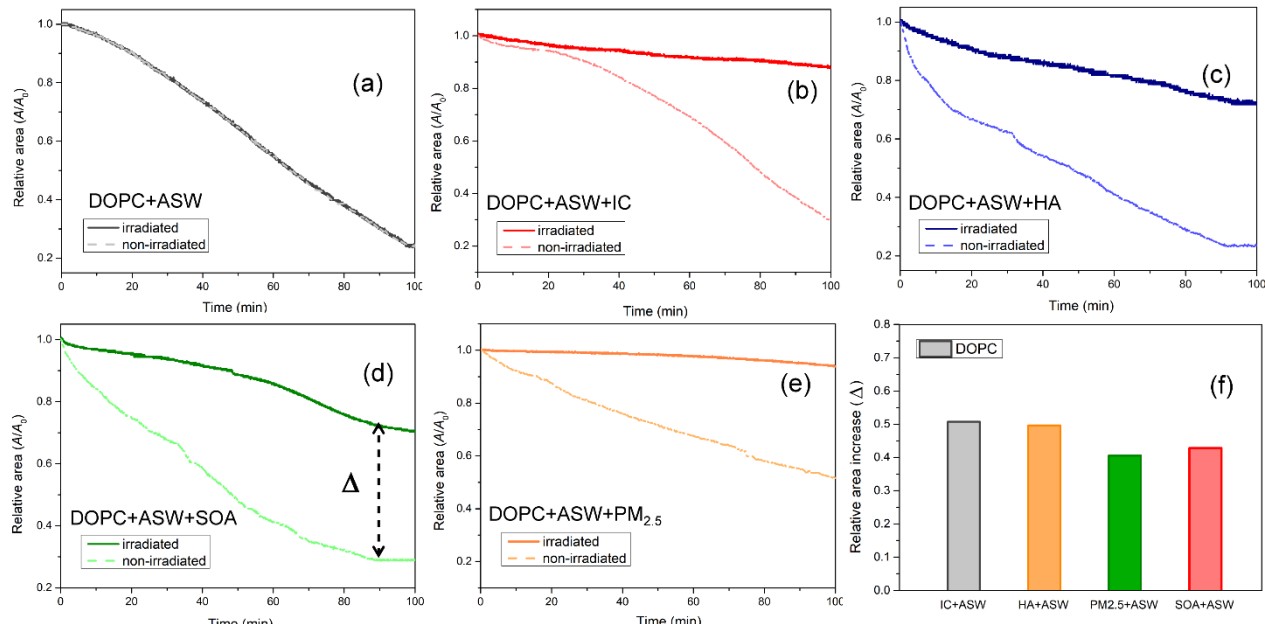

**Figure 4: (a-e) Relative area ($A/A_0$) curves of irradiated and non-irradiated DOPC monolayer on artificial seawater containing IC, HA, PM$_{2.5}$ and SOA samples, respectively; (f) Relative area increase (Δ) of DOPC monolayer after 90 min irradiation; $A_0$ is the molecular area of monolayer at 25 mN/m.**

      In the presence of photosensitizers, the relative areas of the irradiated DOPC monolayers became larger than those of the non-irradiated DOPC as summarized in **Fig. 4(f)**. It is clearly shown where the effect of the photosensitizer on both the DOPC monolayer in the dark and irradiation is illustrated as a relative increase in area (Δ). The presence of IC and HA in the subphase yielded a relative area increase (Δ) of 51% and 50% for DOPC monolayer, respectively. The similar increase of

relative area was also observed in the presence of SOA or PM$_{2.5}$ sample. The relative increases of molecular area for the DOPC monolayer mixed with SOA sample were 43%. The relative area increase for the DOPC monolayer with the PM$_{2.5}$ sample after 90 minutes of irradiation reached 41%. The area loss observed for lipid monolayers for the different compositions of the subphases, at a constant surface pressure, is indicative of their stability (Avila et al., 1999). Compared to the experiments without irradiation, DOPC monolayers were considerably more stable upon irradiation with a photosensitizer. The results of

relative area suggested that photosensitizers induce possible reactions of DOPC under irradiation.

      There were no significant changes of the molecular areas for the DSPC monolayers with and without exposure to light (**Fig. S4-6**). In the irradiation experiments, the decrease of the DSPC monolayer area in artificial seawater with IC and HA is less than 5% after 90 minutes. The loss of the DSPC monolayer area after 90 minutes of irradiation is at least 13%, with respect to the subphase of pure artificial seawater without photosensitizers. In irradiation experiments, the relative areas of the

DSPC monolayer on the artificial seawater mixed with IC or HA were much closer to those experiments without irradiation. Therefore, the stability of the DSPC monolayer on the artificial seawater containing photosensitizers did not change much

between the irradiated and dark experiments. DSPC monolayers are typically more stable than DOPC ones, irrespective of irradiation. The smaller loss of molecular area suggested that the presence of photosensitizer molecules in the subphase improved the stability of the lipid monolayer relative to pure artificial seawater. The different results of relative area curve between the DOPC and DSPC monolayers implied that different reactions were induced by photosensitizers under irradiation.

**3.3 Ordering of lipid monolayers at the air-aqueous interface with or without irradiation**

The molecular-level interactions of photosensitizers with DOPC and DSPC monolayers can be analysed by surface sensitive PM-IRRAS. In the dark experiment on pure artificial seawater (**Fig. 5(a)**) the bands at 2922 and 2853 cm$^{-1}$ were assigned to antisymmetric ($\nu_{as}(CH_2)$) and symmetric methylene (-CH$_2$-) stretching ($\nu_s(CH_2)$) modes, respectively. The antisymmetric ($\nu_{as}(CH_3)$) and symmetric methyl stretching ($\nu_s(CH_3)$) vibrations were observed at 2959 and 2882 cm$^{-1}$,
respectively. The observation of the CH$_3$ bands indicated the *gauche* defects in the alkyl chain (Li et al., 2017a). The CH stretching in the HC=CH group at 3023 cm$^{-1}$ was relatively weak. In the absence of photosensitizers, there were no significant changes of the IRRAS spectra for DOPC monolayer after 90 minutes of irradiation. For the DOPC monolayer on the artificial seawater containing IC molecules, the bands of $\nu_{as}(CH_2)$ and $\nu_s(CH_2)$ stretching were observed at 2923 and 2854 cm$^{-1}$, respectively. After 90 minutes of irradiation, the $\nu_{as}(CH_2)$ and $\nu_s(CH_2)$ bands were shifted to lower wavenumbers at 2921 and
2842 cm$^{-1}$, respectively. The shifts in CH$_2$ and CH$_3$ bands to lower wavenumbers indicate that the *gauche* rotamers in DOPC monolayer were decreased after 90 minutes of irradiation. Therefore, the conformation order of the aliphatic chains in DOPC monolayer was increased. According to **Fig. 5(e)**, the shifts in these CH$_2$ and CH$_3$ bands induced by IC were the most dramatic among the four samples in the photosensitized reaction of organic aerosol surface. The peak height intensity ratio between the antisymmetric CH$_2$ stretching ($I_{as}$) and symmetric CH$_2$ stretching ($I_s$) is usually used to assess the order of the organic
monolayer packing (Aoki et al., 2016; Huang et al., 1982). In the case of IC in the subphase, the peak height intensity ratio between $I_{as}$ and $I_s$ in the DOPC monolayer increased from 1.61 to 1.82 due to irradiation. The increase of peak height ratio also occurred in the presence of HA. It indicated the order of the monolayer chains was increased. With respect to DOPC mixed with the PM$_{2.5}$ sample (**Fig. 5(b)**), the ratio between $I_{as}$ and $I_s$ increased from 1.48 to 1.73 under irradiation. The band of $\nu$(HC=CH) at 3023 cm$^{-1}$ was shifted to 3020 cm$^{-1}$ under the irradiation of the DOPC monolayer mixed with IC. It indicated
that the aliphatic chains became more ordered under irradiation. The existence of the $\nu$(HC=CH) band in the irradiation experiment suggested that the aliphatic chain of DOPC molecules does not break at the initial position of the double bond. The new weak bands at 3001, 3007 and 3009 cm$^{-1}$ that appeared after irradiation was also assigned to $\nu$(HC=CH) stretching. This band implied the formation of unsaturated products in the photosensitized reaction.

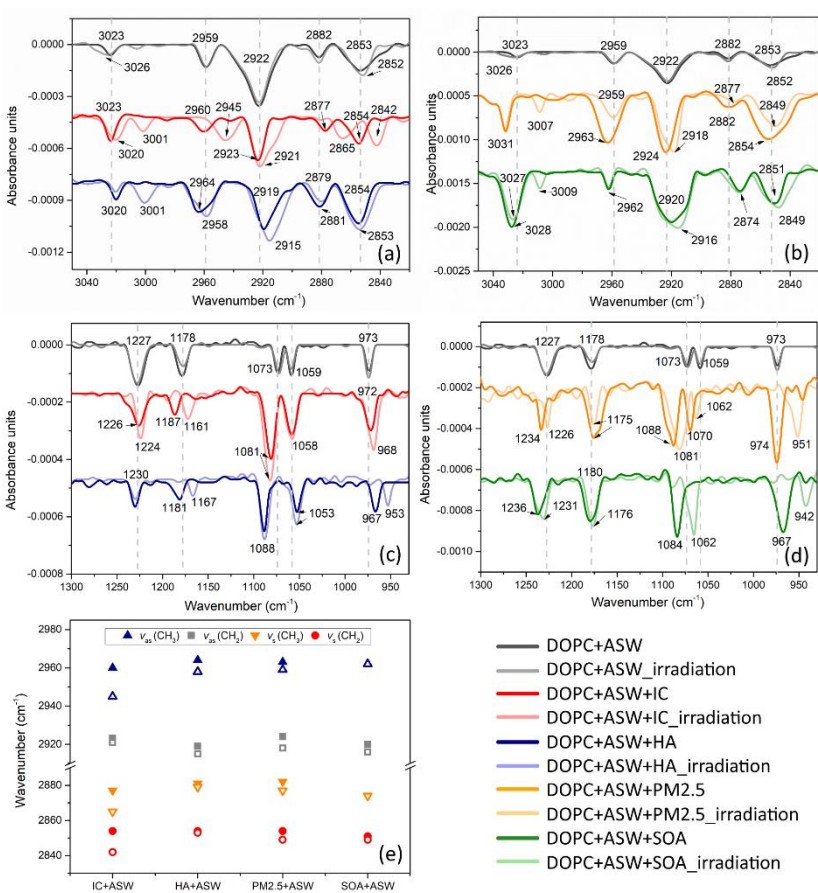

**Figure 5: PM-IRRAS spectra of irradiated and non-irradiated DOPC monolayers in the range of (a) 2820-3050 cm⁻¹ and (c) 930-1300 cm⁻¹ on the artificial seawater containing IC or HA. PM-IRRAS spectra of irradiated and non-irradiated DOPC monolayers in the range of (b) 2820-3050 cm⁻¹ and (d) 930-1300 cm⁻¹ on the artificial seawater containing PM₂.₅ or SOA sample. (e) The frequencies of antisymmetric and symmetric stretching methylene (CH₂) and methyl (CH₃) vibrations for DOPC monolayer on the artificial seawater mixed with IC, HA, PM₂.₅ or SOA sample. IRRAS spectra were recorded at 25 mN/m. (Solid shapes: no irradiation exposure; Open shapes: irradiation exposure)**

The antisymmetric P=O stretching ($\nu_{as}(PO_2^-)$) band and the symmetric stretching ($\nu_s(PO_2^-)$) band for DOPC monolayer on pure artificial seawater (**Fig. 5(c)**) were located at 1227 and 1073 cm⁻¹, respectively. The bands of asymmetric carbonyl ester stretching ($\nu_{as}(CO-O-C)$) at 1187 cm⁻¹ shifted to 1161 and 1167 cm⁻¹ for the DOPC monolayer mixed with IC and HA, respectively. For the DOPC monolayer on the artificial seawater containing SOA sample, the $\nu_{as}(PO_2^-)$ band at 1236 cm⁻¹ and $\nu_s(PO_2^-)$ band at 1084 cm⁻¹ (**Fig. 5(d)**) were shifted to 1231 and 1062 cm⁻¹, respectively. The shifts in the P=O stretching vibrations suggested that hydrogen bonding between phosphate groups and surrounding water molecules appeared to be affected by irradiation. The band at 1059 cm⁻¹ assigned to $\nu_s(C-O-PO_2^-)$ vibration was shifted to 1070 cm⁻¹ for the DOPC

monolayer on the artificial seawater containing PM$_{2.5}$ sample. These shifts in phosphate bands indicated that the interaction between photosensitizer and DOPC molecules induced the dehydration of phosphate groups (Arrondo et al., 1984). In the presence of the SOA sample in the subphase, the antisymmetric stretching of the choline group $v_{as}(CN^+(CH_3)_3)$ band of irradiated DOPC monolayer was shifted from 967 to 942 cm$^{-1}$, which indicated that photosensitizer molecules affected the hydration of DOPC head groups.

The spectral shifts induced by the photosensitizer and irradiation were more obvious for DOPC monolayers than for the DSPC ones. With respect to the DSPC monolayer on the artificial seawater, the antisymmetric CH$_2$ stretching and symmetric CH$_2$ stretching (**Fig. S7(a)**) were 2919 and 2851 cm$^{-1}$, respectively. The CH$_2$ bands in the DSPC monolayers were more intense relative to the DOPC monolayer, thus shifting to lower wavenumbers. This can be attributed to the formation of more a compressed and packed DSPC monolayer relative to DOPC. Lower wavenumbers of CH$_2$ bands are indicative of highly ordered conformation with *all-trans* characteristics (Simon-Kutscher et al., 1996; Christoforou et al., 2012; Snyder et al., 1978). Minimal wavenumber shifts of P=O, C−O−PO$_2^-$ and CN$^+$(CH$_3$)$_3$ stretching vibrations were observed under irradiation in **Fig. S7(b)**. According to the comparison with the IRRAS spectra of DSPC monolayer in dark condition, no new bands of products were observed from the irradiated DSPC film. This result suggested that the DSPC films were less affected by irradiation.

### 3.4 Proposed mechanism for photosensitized reaction of lipid monolayers at the air-aqueous interface

These light-absorbing organic aerosol species can participate in the aging of organic aerosol by acting as photosensitizers. Upon light activation, the excited triplet state of photosensitizer can undergo type I (electron transfer) and/or type II (energy transfer) reactions to generate a variety of highly reactive oxygen species (ROS) (Ding et al., 2011; Aguer et al., 1999). A type I reaction can reduce molecular oxygen (O$_2$), generating oxidizing radicals and radical anion species (e.g., O$_2$•−, HO•), while a type II reaction can transfer energy to O$_2$ to produce singlet oxygen ($^1$O$_2$). The triplet excited state of IC ($^3$IC*), HA ($^3$HA*) and HULIS ($^3$HULIS*) also can directly oxidize other organic material (Tsui et al., 2017; Tsui and McNeill, 2018; Shrestha et al., 2018).

These ROS, along with the triplet excited state of photosensitizers, can react with lipids at the air-aqueous interface, resulting in the photochemical aging of organic film on the aqueous aerosol surface. The slow oxidation of DSPC molecules at the air-sea water interface in the ambient environment does not produce sufficient surface-active molecules, leading to the loss of DSPC molecules from the surface. Previous irradiation studies suggest that green light (530 nm) induced little loss in the surface area of saturated 1,2-dipalmitoyl-sn-glycero-3-phosphocholine (DPPC) monolayer mixed with photosensitive erythrosine or eosin Y molecules (Pereira et al., 2018; Aoki et al., 2016). When considering the liposomal binary mixture of phospholipids containing 1,2-bis(10,12-tricosadiynoyl)-sn-glycero-3-phosphocholine (DC$_{8,9}$PC) and DPPC, DC$_{8,9}$PC can produce intermolecular cross-linking following the release of 5,6-carboxyfluorescein dye under UV irradiation. The same photochemical reaction had no effect on DPPC molecules, even when exposed to radiation with a wavelength of 254 nm, which carries relatively greater energy (Kenaan et al., 2018). Both DSPC and DPPC molecules possess the same

phosphocholine head and saturated aliphatic chain, which is likely why there rather similar activity of DPPC and DSPC with the photosensitizer under irradiation. We inferred that the photo induced reaction did not occur for DSPC monolayer based on the relative area relaxation curves and IRRAS results of DSPC monolayer.

Unsaturated lipids are more susceptible to the attack of $^1O_2$ and free radicals. Reactions of $^1O_2$ with unsaturated bonds in lipid chains can generate hydroperoxide (OOH) groups and then increase molecular area of lipid monolayer due to hydrophilic-hydrophobic balance of the modified chain (Caetano et al., 2007; Riske et al., 2009; Aoki et al., 2016; Pereira et al., 2018). The new band corresponding to CH stretching vibration of HC=CH group was measured by IRRAS, while other bands were changed little in **Fig. 5**. We infer that the structure of photochemical product is similar to DOPC. According to the mechanism of photosensitizing reaction, the unsaturated product detected by IRRAS is possible to be DOPC hydroperoxide. The hydroperoxide group in DOPC is more water soluble than the aliphatic chain of DOPC (**Fig. 6 (a)**). Moreover, DOPC hydroperoxides may further transport toward the hydrophilic region of the monolayer and dissolve into the aqueous phase, leading to a larger surface area of the DOPC monolayer at the air-aqueous interface (Brault et al., 1986). In contrast to pure DOPC monolayer, the formation of hydroperoxides produced by hydroperoxidation reaction between $^1O_2$ and DOPC can be considered as the reason of the distinct relative area increase (Δ) in **Fig. 4(c)**. Giant unilamellar vesicles containing a large amount of unsaturated lipids exhibited a rapid increase in surface area, resulting in morphological changes under irradiation (Riske et al., 2009). Simulations of molecular dynamics also showed that the hydroperoxide group of lipids has a tendency to reside close to the surface with a concomitant expansion of surface area per molecule (Neto and Cordeiro, 2016; Wong-Ekkabut et al., 2007). The PM-IRRAS technique had no response at the OOH band; therefore, we didn't directly detect DOPC hydroperoxides by IRRAS. The new bands of $v$(HC=CH) at 3001, 3007 and 3009 $cm^{-1}$ also support the formation of unsaturated DOPC hydroperoxides present at the air-aqueous interface. The mechanism for the photochemical reaction of the DOPC monolayer by the excited photosensitizer is summarized in **Fig. 6 (b)**. Lipid hydroperoxides are the primary oxidation products and are relatively stable at room temperature. After the initiation phase of hydroperoxide formation, lipid hydroperoxides may further decompose to radicals, leading to the formation of ketones, aldehydes, carboxylic acids, alcohols, esters, and short-chain hydrocarbons (Ghnimi et al., 2017; Choe and Min, 2006). Free radical chain reactions increased the formation of secondary oxidation products. The unsaturated fatty acids also followed this mechanism during the processing of photosensitized oxidation.

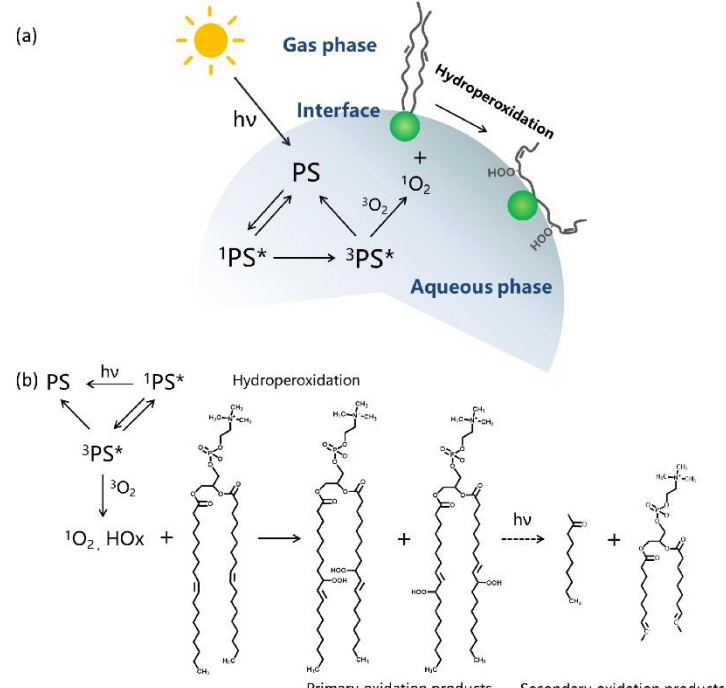

**Figure 6: (a) Proposed model for DOPC-photosensitizer (PS) interaction at the air-aqueous interface. (b) Possible/proposed mechanism for the photo-induced oxidation of DOPC in the presence of photosensitizer (PS).**

Past research has proposed the mechanism for the photosensitized degradation of a saturated fatty acid by the exited photosensitizer at the air-water interface (Tinel et al., 2016; Shrestha et al., 2018; Alpert et al., 2017). The radical produced by the initial hydrogen abstraction of the fatty acid can undergo two possible pathways. In the first pathway, the radical–radical reactions were favoured in an oxygen-poor environment. In the second pathway, in an oxygen-rich environment, the reaction between fatty acid radicals and molecular oxygen can produce peroxy radicals which can further decompose to saturated ketones or other oxidized compounds with hydroxyl and carbonyl functional groups. Then, these compounds can continue to react with molecular oxygen and the excited state of photosensitizer to produce fatty acid radicals and more highly oxidized species.

## 3.5 Atmospheric implication

In this work, photosensitizers like IC and HA in aqueous core of aerosol can take part in the oxidation of unsaturated lipid film coated on the aerosol in the presence of UV light. There was significant organic aerosol aging. The results such as relative area increase and new IRRAS bands of the unsaturated products also indicate that chamber generated SOA samples and authentic $PM_{2.5}$ samples were involved in the photosensitizing reaction of DOPC monolayer. The introduction of authentic

PM$_{2.5}$ sample in this simulated experiment suggested that such process of organic aerosol aging may be occurred in the atmosphere.

The hydrophobic characteristics of organic film were changed greatly in the process of organic aerosol aging. According to the mechanism of photosensitizing reaction, the possible products——DOPC hydroperoxides were more water-soluble. They appear to dissolve into bulk artificial seawater and partition into the hydrophilic core of organic aqueous aerosols. Subsequently, the processes of hygroscopic growth of aerosol and cloud condensation nuclei activation are impacted. If the organic film on the aerosol surface is either destabilized under irradiation or it is metastable due to the loss of ordering and packing, the organic aerosol will become more permeable to water. As a result, the hygroscopicity of the aerosol particle and the overall size of droplet can increase (Ruehl and Wilson, 2014). Consequently, the atmospheric lifetime of unsaturated species on the aerosol surface can decrease. Volatility is generally inversely correlated with O:C ratio (Aiken et al., 2008). The photochemical reaction at the air-aqueous interface under the condition of BrC is an efficient and common pathway to oxidize organic films toward low volatile organic compounds (O:C ratio of 0.25 to 1) (Jimenez et al., 2009). Moreover, the photosensitized reaction of organic aqueous aerosol likely depends on the film-forming species, given the different reactivities of saturated and unsaturated phospholipids.

## 4 Conclusions

We have investigated the photosensitized reactions of saturated and unsaturated organic monolayers at the air–water interface as proxies for the photochemical aging of organic-coated aqueous aerosols by BrC. The surfactant molecules SA, EA, OA, DSPC and DOPC were chosen to investigate the effects of degree of unsaturation and conformation on the ordering, packing and stability of organic film. The surface areas of the unsaturated lipids (EA, OA and DOPC) with the double bond in the 9-position were much more expanded than the saturated conformations with the same chain lengths. The type of configurational isomers (*cis* and *trans*) affected the surface properties of the organic monolayer. The π–A isotherms suggested that the EA monolayer had a slightly higher compressibility than the OA monolayer. According to relative area increase curves, the addition of photosensitizers into the subphase increased the stability of DSPC and DOPC monolayers on the aerosol surface, irrespective of irradiation or non-irradiation. The saturated DSPC monolayer has greater photochemical stability than the DOPC monolayer. The relative areas of DOPC monolayers in the artificial seawater containing photosensitizers were increased after irradiation. In addition, the largest increase of the relative area of the DOPC monolayer was observed in the presence of IC, as compared to the laboratory generated SOA sample and field collected PM$_{2.5}$ sample. The changes between the irradiated and dark PM-IRRAS spectra also support the photochemical oxidation of the unsaturated chains in DOPC. DOPC hydroperoxides were considered to be primary oxidation products during the photochemical oxidation of unsaturated lipids via singlet oxygen generation. The photosensitized reaction of DOPC monolayers at the surface of aqueous aerosol can lead to the formation of more oxidized and therefore less reactive species at the surface. Overall, the transformation of hydrophobic organic materials into hydrophilic organic components can alter the hygroscopicity of an organic aerosol particle.

The Supplement related to this article is available online.

**Author contributions**

LD and SL conceived and led the studies. SL, XJ and LL carried out the experiments and analysed the data. SL, LD, CG and MR interpreted the results. WJ, QZ, WW and MG discussed the results and commented on the manuscript. SL prepared the manuscript with contributions from all co-authors.

**Competing interests**

The authors declare that they have no conflict of interest.

**Acknowledgements**

This work was supported by National Natural Science Foundation of China (91644214, 21876098), Shandong Natural Science Fund for Distinguished Young Scholars (JQ201705), Shandong Key R&D Program (2018GSF117040), Fundamental Research Funds of Shandong University (2017JQ01), Taishan Scholars (ts201712003) and Marie Curie International Research Staff Exchange project MARSU (Grant 690958).

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
