# Peer review of "Supporting information"

_Atmospheric Chemistry and Physics, 2019_

## Referee Comment (RC1) · Anonymous Referee #1 · 14 Feb 2019

The work of Li et al. presents a detailed investigation of photochemical aging of atmospheric relevant organic films at the air-aqueous interface. The stability and oxidized products of organic film under UV irradiation were studied with the aid of Langmuir trough and Infrared reflection absorption spectroscopy (IRRAS). This work represents a significant advance, as the water soluble organic compounds extracted from atmospheric and chamber samples were used in the Langmuir experiments for the first time. These methods play an actual role in researching the photochemical aging of organic-coated aqueous aerosols, and the further atmospheric implications. The area expansion of DOPC film was revealed by relaxation curves and was further confirmed by IRRAS spectra. The authors illustrated the mechanisms for the photosensitizing

reaction of organic film consisted by unsaturated lipid and brown carbon. The authors have established a representative model for organic aerosol coating with advanced instruments. The methods are appropriate and properly conducted, and the experimental data presented in this work is of high quality. The manuscript is well written, although the authors need to make some small modifications (see details below). This manuscript can be published in Atmospheric Chemistry and Physics after revisions in consideration of the following comments:

Page 4: Why did the authors mix photosensitizers with the artificial seawater? The effect of artificial seawater in the experiment should be described.

Page 5, line 1: The chamber experiment needs to be more specific on the drying conditions.

Page 7, line 14: The surface state of DOPC monolayers underwent a transition from the gas phase to the liquid phase. The authors should describe gas and liquid phase of monolayer.

Page 12: The y-axes in Figure 5 that are being directly compared had different ranges. It might make more sense to compare A and C in the same ranges (so, switch the y-axes for subplots C and D).

Page 16: Organic coatings on the aerosols are also important for multiphase chemistry in the atmosphere (i.e. N2O5, HNO3 uptake). A more expansive discussion of the effect of organic film on the multiphase aerosol chemistry in the atmospheric implication section is encouraged.

---

## Referee Comment (RC2) · Anonymous Referee #2 · 4 Apr 2019

Li and co-authors examined the packing of surfactants at the air-water interface, how this varies with different surfactants, and how it changes with irradiation in the presence of four types of photosensitizers. They also examined how irradiation changes the properties of the films. While there are some components here that are interesting, the environmental implications of the work are not always clear. For example, how can the pi-A isotherm figures help us understand something about atmospheric particles? I am also concerned that the interpretation of the irradiation data for DOPC, which was the focus of the illumination experiments, is confounded by the fast dark reaction of this unsaturated phospholipid. Overall, I would consider this manuscript to straddle the border of reject/major revisions.

[Figure]

»Major points

**Photosensitizer issues** The paper makes many comparisons between the relative effects of the four sensitizers (IC, humic acids, limonene SOA, and ambient PM). But these effects likely depend on the concentrations of the sensitizers, which were different (and apparently arbitrary) for the four sensitizers. Based on this, it seems that the sensitizer comparisons are meaningless. (For example, see line 13-14 in the Conclusions: ". . . IC was the most efficient photosensitizer to increase the relative area of the DOPC monolayer. . .".)

Also, the concentrations of photosensitizers seem quite high: how do they compare to atmospherically relevant amounts in airborne particles with typical liquid water contents? The IC concentration (2.5 mM) seems especially high since it appears to be of intermediate volatility and would primarily partition to the gas phase in an aerosol. This raises a question: is the impact of a photosensitizer proportional to its concentration? For example, in Fig. 3, is the influence of IC on the pi-A isotherms proportional to IC concentration?

The aqueous mass concentrations (mg-PM/L-solution) of the PM2.5 and SOA samples in ASW are not given: these need to be included.

It would be helpful to show a figure with UV/Vis spectra of the four sensitizers at the concentrations used in the experiments. At least this would allow the reader to understand the differences in the rate of light absorption in the four cases, as this would influence the formation of singlet oxygen.

**Irradiation issues** Fig. 4. DOPC in artificial seawater (ASW) has a short half-life, approximately 70 min, both in the dark and under irradiation (Fig. S1). The authors attribute this rapid DOPC loss to reactions with gas-phase oxidants. Compared to the ASW base case, the loss of "relative area" is slowed in irradiated samples containing a photosensitizer, which the authors attribute to formation of hydroperoxides, but they have not analyzed for this functional group. Fundamentally, the relative area (A/A0)

measure of Fig. 4 is very crude and the interpretation of the results is very poorly constrained. For example, it seems possible that the photosensitizer could reduce the loss of DOPC by making products that slow DOPC oxidation by gas-phase oxidants. Or a larger product other than hydroperoxides could be made by the interaction of sensitizer and DOPC. The authors need better evidence for their interpretation; this should start by doing the irradiation in a sealed container so that gas-phase oxidants are not rapidly destroying DOPC.

What is the most important point from these results? If it is that hydroperoxides are formed, then peroxides should be analyzed. If it is that the DOPC products have larger molecular areas than DOPC, then the rapid background loss of relative area needs to be stopped.

**Section 3.3. PM-IRRAS results** This section of text, currently 3 pages, is too long, is very dry and is too focused on the details of various band assignments. Similarly, Figures 5 and 6 generally show only very subtle differences between some of the bands after irradiation. Much of this section could be moved to the supplemental material so that the main text contains a 1-page summary that focuses on the most important results.

**Section 3.5. Atmospheric implication** This section should focus less on a review of what others have done and more on the implications of the current work. What do the current results tell us that we didn't know before? The second sentence of this section states that salt particles are covered with a film of surfactants, but as I understand it, this is still a topic of debate. Similarly, I believe there is debate about whether an organic film on particles is an effective barrier to mass transport, e.g., of water vapor, as this section states. Given that the photosensitizer concentrations were very high in the current work, can a timescale for oxidation under atmospheric conditions be estimated?

»Other points page 4, line 19: Sonication is a poor choice to remove PM from filters

because it can oxidize organics. What was the power of the ultrasonic bath? How long were samples sonicated?

page 5, top: Need more details on the chamber experimental conditions, including a supplemental table describing different chamber experiments. What were concentrations of H2O2 and NO in the chamber? How long was the reaction allowed to proceed before particles were collected? What was the concentration of limonene that was reacted? What was SOA mass collected? Were DLPI stages combined to get one PM extract per chamber experiment?

page 6. It would be very helpful to give a short description of how a pi-A isotherm can be interpreted. I imagine most readers, like myself, are not familiar with reading these types of figures. What information does the isotherm reveal? What is a lift-off area? What is a collapse? How are these determined from the isotherm? Why are these quantities important? Amending Figure 1 to show a molecular picture of the various stages in the pi-A isotherm would help.

page 9, line 12. Indicate that this is 30% after 90 min of irradiation.

lines 12 – 14: "There was evidently.." This statement is contrary to the data: the addition of photosensitizer appears to decrease the decay of the DOPC monolayer. This sentence is then contradicted by the next sentence ("The presence of . . .").

»Minor points page 1, lines 20-22: Define OA and EA. Also, the sentence is unclear. What is the comparison? line 24: Since there is no direct experimental evidence for hydroperoxidation in the current work, this statement should be qualified. line 28: "the processing of organic aerosol aging" does not "control" aerosol composition.

page 2, l.28: This is poorly worded: the triplet state is not susceptible to oxidation by a hydrocarbon.

page 5, line 20: Is TUV the model number of the lights? If not, what is model number? What was the photon flux in the sample?

Figure 1: The structures are very small and difficult to discern, especially the double bonds.

page 6, line 17: This sentence is not precise enough: the entire DOPC molecule didn't go from gas phase to aqueous phase.

page 7, line 5: "in both the liquid and condensed phase". How is "condensed" phase different from "liquid" phase?

line 14: "for the DPPC monolayer". Shouldn't this be DOPC?

line 17: "The introduction of photosensitizers...had a profound effect...". This is only true for HA, not for IC.

---

## Referee Comment (RC3) · Anonymous Referee #3 · 9 Apr 2019

The comment was uploaded in the form of a supplement: https://www.atmos-chem-phys-discuss.net/acp-2019-96/acp-2019-96-RC3-supplement.pdf
* * *

---

## Author Comment (AC1) · 3 Jun 2019

We have revised our manuscript according to the suggestions of the Referee's comments and the responses to the comments are as following. For clarity, the Referee's comments are reproduced in blue, authors' responses are in black and changes in the manuscript are in red color text.

**Anonymous Referee #1**
The work of Li et al. presents a detailed investigation of photochemical aging of atmospheric relevant organic films at the air-aqueous interface. The stability and oxidized products of organic film under UV irradiation were studied with the aid of Langmuir trough and Infrared reflection absorption spectroscopy (IRRAS). This work represents a significant advance, as the water soluble organic compounds extracted from atmospheric and chamber samples were used in the Langmuir experiments for the first time. These methods play an actual role in researching the photochemical aging of organic-coated aqueous aerosols, and the further atmospheric implications. The area expansion of DOPC film was revealed by relaxation curves and was further confirmed by IRRAS spectra. The authors illustrated the mechanisms for the photosensitizing reaction of organic film consisted by unsaturated lipid and brown carbon. The authors have established a representative model for organic aerosol coating with advanced instruments. The methods are appropriate and properly conducted, and the experimental data presented in this work is of high quality. The manuscript is well written, although the authors need to make some small modifications (see details below). This manuscript can be published in Atmospheric Chemistry and Physics after revisions in consideration of the following comments:

Page 4: Why did the authors mix photosensitizers with the artificial seawater? The effect of artificial seawater in the experiment should be described.

Response:

The components of artificial seawater are approximately equal to fresh sea spray. According to our previous studies, artificial sea salts also play a role in stabilizing the organic films (*Atmos. Environ.*, **2019**, 200, 15–23; *Environ. Pollut.*, **2018**, 242, 626–633). We have added some description about the effect of artificial seawater in the experimental section on Page 4:

"The effect of artificial seawater on stabilizing organic monolayer has been confirmed in previous studies (Li et al., 2018; Li et al., 2019)."

Page 5, line 1: The chamber experiment needs to be more specific on the drying conditions.

Response:

The chamber experiment was performed at RH 20%, which is lower than the crystallization RH (35%) of ammonium sulfate (*Atmos. Chem. Phys.*, **2007**, 7, 3909–3922). The seed particle was kept at solid phase.

We have added some discussion on Page 5:

"Relative humidity was about 20% throughout the experiments, which is lower than the crystallization RH (35%) of $(NH_4)_2SO_4$ (Ng et al., 2007). The seed particle was kept at solid phase."

 The surface state of DOPC monolayers underwent a transition from the gas phase to the liquid phase. The authors should describe gas and liquid phase of monolayer.

Response:

Surface pressure–area isotherms show phase transition-like behaviour of the Langmuir films. In the gas phase, there is minimal pressure increase for a decrease in area. The first transition from gas phase to liquid phase occurs at the lift-off area. There is a proportional increase in surface pressure with decreasing area. Moving into the solid region of monolayer is accompanied by another sharp transition to a more severe area dependent pressure. This trend continues up to a point where the molecules are packed closely and have very little space to move. Applying an increasing pressure at this point causes the monolayer to become unstable and destroy the monolayer. We have added some description of phase transitions and the insert of monolayer behaviours at the various stages in the surface pressure-area isotherm on Page 8:

"When the surface area of the DOPC monolayer was larger than 125 Å$^2$/molecule, the distance between the DOPC molecules was quite large and the intermolecular force was quite weak. The surface pressure of a DOPC monolayer on artificial seawater started to increase from 125 Å$^2$/molecule, where the DOPC monolayer surface state underwent a transition from the gas to the liquid-condensed phase. After the first phase transition, there is a proportional increase in surface pressure with decreasing area. This caused condensation and ordering at the interface, increasing the surface pressure of the organic monolayer. This trend continued up to a point where the DOPC molecules were packed closely and have very little space to move. Finally, the DOPC monolayer collapsed at 46 mN/m (Pereira et al., 2018). Applying an increasing pressure at the collapse pressure caused the monolayer to become unstable and destroy the monolayer. The packing and phase behaviours of DOPC monolayer at different phases were also shown in the inserts in **Fig. 2(a)**."

 The y-axes in Figure 5 that are being directly compared had different ranges. It might make more sense to compare A and C in the same ranges (so, switch the y-axes for subplots C and D).

Response:

We have switched the y-axes for A and B in Figure 5.

 Organic coatings on the aerosols are also important for multiphase chemistry in the atmosphere (i.e. N2O5, HNO3 uptake). A more expansive discussion of the effect of organic film on the multiphase aerosol chemistry in the atmospheric implication section is encouraged.

Response:

We have added some discussion about the effect of organic film on the multiphase aerosol chemistry on Page 3:

"The packing order and stability of films on the surface also facilitate further discussion of the uptake efficiency of marine aerosols toward atmospheric trace gases such as $N_2O_5$

and $HNO_3$ (Bertram et al., 2018)."

---

## Author Comment (AC2) · 3 Jun 2019

We have revised our manuscript according to the suggestions of the Referee's comments and the responses to the comments are as following. For clarity, the Referee's comments are reproduced in blue, authors' responses are in black and changes in the manuscript are in red color text.

**Anonymous Referee #2**
Li and co-authors examined the packing of surfactants at the air-water interface, how this varies with different surfactants, and how it changes with irradiation in the presence of four types of photosensitizers. They also examined how irradiation changes the properties of the films. While there are some components here that are interesting, the environmental implications of the work are not always clear. For example, how can the pi-A isotherm figures help us understand something about atmospheric particles? I am also concerned that the interpretation of the irradiation data for DOPC, which was the focus of the illumination experiments, is confounded by the fast dark reaction of this unsaturated phospholipid. Overall, I would consider this manuscript to straddle the border of reject/major revisions.

»Major points
**Photosensitizer issues** The paper makes many comparisons between the relative effects of the four sensitizers (IC, humic acids, limonene SOA, and ambient PM). But these effects likely depend on the concentrations of the sensitizers, which were different (and apparently arbitrary) for the four sensitizers. Based on this, it seems that the sensitizer comparisons are meaningless. (For example, see line 13-14 in the Conclusions: "…IC was the most efficient photosensitizer to increase the relative area of the DOPC monolayer…".)

Response:

We have deleted the discussion about the comparison of the photosensitizing efficiency of the four photosensitizers at different concentrations. The change of relative area of DOPC monolayer in the artificial seawater containing IC was the greatest. The photosensitizing reaction of unsaturated lipids involving $PM_{2.5}$ sample and SOA sample indicated that such reaction may occur in the ambient environment. We have modified the statement on Page 13 and 18:

"According to **Fig. 5(e)**, the shifts in these $CH_2$ and $CH_3$ bands induced by IC were the most dramatic among the four samples in the photosensitized reaction of organic aerosol surface."

"In addition, the largest increase of the relative area of the DOPC monolayer was observed in the presence of IC, as compared to the laboratory generated SOA sample and field collected $PM_{2.5}$ sample."

Also, the concentrations of photosensitizers seem quite high: how do they compare to atmospherically relevant amounts in airborne particles with typical liquid water contents? The IC concentration (2.5 mM) seems especially high since it appears to be of intermediate volatility and would primarily partition to the gas phase in an aerosol. This raises a question: is the impact of a photosensitizer proportional to its

concentration? For example, in Fig. 3, is the influence of IC on the pi-A isotherms proportional to IC concentration?

Response:

The concentration of 30 mg/L humic acid is commonly used in the photochemical experiment (*Environ. Sci. Technol.*, **2015**, 49, 13199-13205; *Sci. Rep.*, **2015**, 5, 12741). In the previous studies, larger than or equal to 30 mg/L humic acid were added directly into the artificial seawater to mimic the presence of the dissolved organic matter in the sea surface microlayer (*Sci. Rep.*, **2015**, 5, 12741; *Geophys. Res. Lett.*, **2017**, 44, 1079–1087). In this investigation, the concentration of humic acid in fresh sea spray is close to the real seawater.

These dissolved organic matters are present in different and varying concentrations in seawater and aerosols. The concentrations of IC commonly used in the experiment varied from 0.25 mM to 0.6 M (*Environ. Sci. Technol.*, **2018**, 52, 7680−7688; *Faraday Discuss.*, **2013**, 165, 123–134; *C. R. Chimie*, **2014**, 17, 801–807). IC can also be chosen as seed particle in some chamber simulation (*Environ. Sci. Technol.*, **2014**, 48, 6, 3218–3227). The concentration of IC used in our experiment is lower than its saturated concentration. The surface activity of organic film gives rise to a concentrating effect at the interface. Therefore, the chromophoric dissolved organic matters are more concentrated in the surface of sea microlayer than in the bulk seawater. These organic compounds may be present in the aerosol at much higher concentrations than in either the seawater or air bulk phases. In term of photosensitizing efficiency, the impact of a photosensitizer is dependent on its concentration. We have deleted the discussion about the comparison of their photosensitizing efficiency at different concentrations.

The aqueous mass concentrations (mg-PM/L-solution) of the PM2.5 and SOA samples in ASW are not given: these need to be included.

Response:

In the revised manuscript, we have given the aqueous mass concentration of $PM_{2.5}$ and SOA samples, and added some description about this issue on Page 5-6:

"All the filters were dissolved into 40 mL ultrapure water with ultrasonic agitation. Sonication was performed in an ultrasonic bath with a frequency of 40 kHz and the power of 80 W. The sonication time was 15 min. Subsequently, the suspension was centrifuged at 1780 g for 40 min. The supernatant, which contains the water-soluble fraction including water-soluble organic compounds (WSOC) and inorganic ions, was re-collected by freeze-drying. The insoluble fraction separated from soluble fraction was also freeze-dried. The mass ratio of insoluble to soluble fraction is 0.91:1. Then, 3.3 mg of freeze-dried soluble sample was dissolved in 1000 mL artificial seawater. The concentration of $PM_{2.5}$ sample in the artificial seawater is 3.3 mg/L."

"Then, the SOA samples collected on the aluminium foil pieces were dissolved in ultrapure water by sonicating for 1 min in an ultrasonic bath. The extract water solution was concentrated by rotary evaporation. The residue was dried under high purity nitrogen stream. Then, the SOA sample was transferred to the artificial seawater with the concentration of 0.66 mg/L."

It would be helpful to show a figure with UV/Vis spectra of the four sensitizers at the concentrations used in the experiments. At least this would allow the reader to understand the differences in the rate of light absorption in the four cases, as this would influence the formation of singlet oxygen.

Response:

UV–vis spectra were acquired on a UV–vis spectrophotometer (P9, Shanghai Mapada) using 1 cm quartz cuvettes. The absorption spectra of the aqueous solutions for 0.006 g/L IC, 0.006 g/L humic acid, 0.01 g/L $PM_{2.5}$ and 0.01 g/L SOA were recorded. In order to compare the light-absorbing properties of four samples, we also calculate MAC and average MAC (<MAC>) values over the wavelengths range from $\lambda_1$ = 200 nm to $\lambda_n$ = 700 nm). We have added the UV-vis spectra measurements in the experimental section on Page 6:

"The UV–vis absorption spectra of the four photosensitizers were measured using a UV-vis spectrophotometer (P9, Shanghai Mapada, China). Spectra were collected using quartz cuvettes with internal path length of 1.0 cm. Aqueous solutions of 0.006 g/L IC, 0.006 g/L humic acid, 0.01 g/L $PM_{2.5}$ sample and 0.01 g/L SOA sample were used. The IC aqueous solution displays a major absorption band at 288 nm which is in an agreement with previous studies (Tinel et al., 2014; Berke et al., 2019). The maximum absorption of SOA sample was at 286 nm, as can be seen from the absorption spectrum shown in Fig. S1 in the supplement. Similar to the previous studies, HA and $PM_{2.5}$ samples' absorptions have the main feature around 200 nm with a clearly visible hump between 250 and 300 nm (Kristensen et al., 2015). We calculate the mass-absorption coefficients (MAC) ($cm^2/g$) of the four samples by the following equation:

$$MAC(\lambda) = \frac{A(\lambda) \times \ln 10}{b \times C_m}, \qquad (1)$$

where A is the absorption, b is the length of the cuvette (1 cm), and $C_m$ is the concentration of the dissolved reaction products (g/mL).

In order to compare the light-absorbing properties of the four samples, we estimated the light-absorbing properties of aerosols by the average MAC (<MAC>) over the wavelengths range from $\lambda_1$ = 200 nm to $\lambda_n$ = 700 nm) (Jiang et al., 2019).

$$\langle MAC \rangle = \frac{\sum_{i=1}^{n} MAC(\lambda_i)}{n}, \qquad (2)$$

HA sample with <MAC>$_{200–700\ nm}$ of 64460 $cm^2/g$ is also more absorbing than IC sample (<MAC>$_{200–700\ nm}$ = 41267 $cm^2/g$), SOA sample (<MAC>$_{200–700\ nm}$ = 32867 $cm^2/g$) and $PM_{2.5}$ sample (<MAC>$_{200–700\ nm}$ = 16048 $cm^2/g$)."

**Irradiation issues** Fig. 4. DOPC in artificial seawater (ASW) has a short half-life, approximately 70 min, both in the dark and under irradiation (Fig. S1). The authors attribute this rapid DOPC loss to reactions with gas-phase oxidants. Compared to the ASW base case, the loss of "relative area" is slowed in irradiated samples containing a photosensitizer, which the authors attribute to formation of hydroperoxides, but they

have not analyzed for this functional group. Fundamentally, the relative area (A/A$_0$) measure of Fig. 4 is very crude and the interpretation of the results is very poorly constrained. For example, it seems possible that the photosensitizer could reduce the loss of DOPC by making products that slow DOPC oxidation by gas-phase oxidants. Or a larger product other than hydroperoxides could be made by the interaction of sensitizer and DOPC. The authors need better evidence for their interpretation; this should start by doing the irradiation in a sealed container so that gas-phase oxidants are not rapidly destroying DOPC.

Response:

To lower the content of reactive species in the atmosphere adjacent to the film, the trough was placed in a sealed box in some experiments. The DOPC monolayers in the artificial seawater displayed a trend for decreasing area, probably owing to loss of material to the subphase (*Langmuir*, **2016**, 32, 3766−3773; *Biophys. J.*, **2010**, 98, 50-52). At a relative area of approximately 0.18, the barriers could not move anymore and the experiment was terminated.

Indeed, it may be a possible explanation for the increase of relative area that the photosensitizer could reduce the loss of DOPC by making products which slow DOPC oxidation by gas-phase oxidants. However, the existence of new bands at approximately 3000 cm$^{-1}$ suggested that DOPC and photosensitizers under irradiation can produce some unsaturated organic compounds. According to the mechanism of photosensitizing reaction, DOPC hydroperoxides are the primary products (*Langmuir*, **2007**, 23, 1307-1314; *Biophys. J.*, **2009**, 97, 1362-1370; *Colloid Surface B*, **2018**, 171, 682-689). Additionally, no other new bands but the shifts of bands for DOPC were measured by IRRAS. Though IRRAS spectra are quite indistinct in the range of 3500-4000 cm$^{-1}$ for OH stretching vibration, the unsaturated products measured by IRRAS can still be assigned as DOPC hydroperoxide.

The irradiation experiments also can be performed in a completely sealed container that was purged with nitrogen to protect unsaturated lipids from oxidation. Such experiments may not accord with the ambient environment. Moreover, DOPC needs to be exposed to oxygen in the photosensitizing reaction. Therefore, the irradiation in a sealed container was not considered in the experiment.

What is the most important point from these results? If it is that hydroperoxides are formed, then peroxides should be analyzed. If it is that the DOPC products have larger molecular areas than DOPC, then the rapid background loss of relative area needs to be stopped.

Response:

The new band corresponding to CH stretching vibration of HC=CH group was measured by IRRAS, while other bands were changed little. Accordingly, the structure of photochemical product must be similar to DOPC. The photosensitizing reaction mechanism of unsaturated lipids proposed here was generally accepted (*Langmuir*, **2016**, 32, 3766-3773; *Langmuir*, **2007**, 23, 1307-1314; *Biophys. J.*, **2009**, 97, 1362-1370; *Colloid Surface B*, **2018**, 171, 682-689). Based on the increase of relative area and the detected products, we tentatively assigned the unsaturated products measured

by IRRAS as primary oxidation products——DOPC hydroperoxide.

**Section 3.3. PM-IRRAS results** This section of text, currently 3 pages, is too long, is very dry and is too focused on the details of various band assignments. Similarly, Figures 5 and 6 generally show only very subtle differences between some of the bands after irradiation. Much of this section could be moved to the supplemental material so that the main text contains a 1-page summary that focuses on the most important results.
Response:
Figure 6 for DSPC monolayer has been moved to the supplement. We have shortened this section significantly on Page 13:

[revised manuscript text omitted]

**Section 3.5. Atmospheric implication** This section should focus less on a review of what others have done and more on the implications of the current work. What do the current results tell us that we didn't know before? The second sentence of this section states that salt particles are covered with a film of surfactants, but as I understand it, this is still a topic of debate. Similarly, I believe there is debate about whether an organic film on particles is an effective barrier to mass transport, e.g., of water vapor, as this section states. Given that the photosensitizer concentrations were very high in the current work, can a timescale for oxidation under atmospheric conditions be estimated?

Response:

We have moved this material to the introduction section and modified the atmospheric implication section on Page 17-18:

"In this work, photosensitizers like IC and HA in aqueous core of aerosol can take part

in the oxidation of unsaturated lipid film coated on the aerosol in the presence of UV light. There was significant organic aerosol aging. The results such as relative area increase and new IRRAS bands of the unsaturated products also indicate that chamber generated SOA samples and authentic $PM_{2.5}$ samples were involved in the photosensitizing reaction of DOPC monolayer. The introduction of authentic $PM_{2.5}$ sample in this simulated experiment suggested that such process of organic aerosol aging may be occurred in the atmosphere.

The hydrophobic characteristics of organic film were changed greatly in the process of organic aerosol aging. According to the mechanism of photosensitizing reaction, the possible products——DOPC hydroperoxides were more water-soluble. They appear to dissolve into bulk artificial seawater and partition into the hydrophilic core of organic aqueous aerosols. Subsequently, the processes of hygroscopic growth of aerosol and cloud condensation nuclei activation are impacted. If the organic film on the aerosol surface is either destabilized under irradiation or it is metastable due to the loss of ordering and packing, the organic aerosol will become more permeable to water. As a result, the hygroscopicity of the aerosol particle and the overall size of droplet can increase (Ruehl and Wilson, 2014). Consequently, the atmospheric lifetime of unsaturated species on the aerosol surface can decrease. Volatility is generally inversely correlated with O:C ratio (Aiken et al., 2008). The photochemical reaction at the air-aqueous interface under the condition of BrC is an efficient and common pathway to oxidize organic films toward low volatile organic compounds (O:C ratio of 0.25 to 1) (Jimenez et al., 2009). Moreover, the photosensitized reaction of organic aqueous aerosol likely depends on the film-forming species, given the different reactivities of saturated and unsaturated phospholipids."

»Other points page 4, line 19: Sonication is a poor choice to remove PM from filters because it can oxidize organics. What was the power of the ultrasonic bath? How long were samples sonicated?

Response:

It has been an increased interest in the use of ultrasound to destroy organic pollutants. Long time sonication with methanol or water may generate OH radicals to oxidize organics (*J. Phys. Chem.*, **1983**, 87, 1369–1377). The filter sample was extracted by sonication with 40 mL ultrapure water for 15 min. The rated frequency of ultrasound is 40 kHz. Higher frequency ultrasound can increase the number of free radicals in the system.

We have added some explanation of sonication on Page 5:

"All the filters were dissolved into 40 mL ultrapure water with ultrasonic agitation. Sonication was performed in an ultrasonic bath with a frequency of 40 kHz and the power of 80 W. The sonication time was 15 min."

page 5, top: Need more details on the chamber experimental conditions, including a supplemental table describing different chamber experiments. What were concentrations of $H_2O_2$ and NO in the chamber? How long was the reaction allowed to proceed before particles were collected? What was the concentration of limonene that

was reacted? What was SOA mass collected? Were DLPI stages combined to get one PM extract per chamber experiment?

Response:

Temperature, relative humidity (RH), $O_3$, NO, and $NO_x$ were continuously monitored. $H_2O_2$ was used as OH radical precursor and was introduced into the chamber by passing pure zero air over 20 μL $H_2O_2$ (Sigma-Aldrich, 30 wt% in $H_2O$) solution. The concentration of $H_2O_2$ was estimated to be 4324 ppb. RH was about 20 % throughout the experiments. The concentration of OH radicals in the chamber cannot be explicitly determined due to lack of appropriate device. The concentration of limonene was determined by gas chromatograph equipped with flame ionization detector (GC-FID) (Agilent Technologies, GC-FID 7890B). The chromatographic separation was achieved by using a DB-624 capillary column (Agilent Technologies, 30 m length × 1.8 μm film thickness × 0.32 mm i.d.). The GC oven temperature was heated at a rate of 2 ℃/min from 180 ℃ to 186 ℃. At this temperature, the peaks for each reference would not overlap with the reactant. The relative concentrations of each compound were determined from peak areas. The chamber was flushed using zero air three times after each experiment. The reaction was allowed to proceed for 4 hours before SOA particles were collected. At that time, the concentration of limonene could not be detected. The initial concentration of $NO_x$ and limonene were 206 ppb and 684 ppb, respectively. The SOA mass was 0.22 mg. The aluminium foil pieces collected from each stage were all combined to get one PM extract. The experimental conditions of chamber were also listed in Table S1 in the supplement.

We have added more details on Page 5:

"A customized diffusion dryer was added after aerosol generator to make sure that the $(NH_4)_2SO_4$ aerosols were in solid phase in chamber. Relative humidity was about 20% throughout the experiments, which is lower than the crystallization RH (35%) of $(NH_4)_2SO_4$ (Ng et al., 2007). The seed particle was kept at solid phase. The total gas volume in the chamber was recorded with mass flow meters (D80-8C/ZM, Beijing Sevenstar, China). Limonene (99%, tci) was injected into the chamber by a micro syringe and was evaporated into a stream of purified air. Then, an aqueous $H_2O_2$ solution (30 wt %) was injected to the chamber and served as the OH precursor in these experiments. The concentration of $H_2O_2$ was estimated to be 4324 ppb. NO was introduced into the chamber by a gas-tight syringe. Typically, 684 ppb limonene and $5×10^4$ $cm^{-3}$ $(NH_4)_2SO_4$ seed aerosols were employed. The concentration of limonene was determined by gas chromatograph equipped with flame ionization detector (GC-FID) (Agilent Technologies, GC-FID 7890B). The SOA formation of limonene photooxidation experiments was performed under high-$NO_x$ condition (Sarrafzadeh et al., 2016). The initial concentration of $NO_x$ and NO detected by $NO-NO_2-NO_x$ analyzer (Model 42C, Thermo Electron Corporation, USA) were 206 ppb and 164 ppb, respectively. The reaction was allowed to proceed for 4 hours before SOA particles were collected. At that time, the concentration of limonene could not be detected. The initial concentrations of reactants in the chamber were also listed in Table S1 in the supplement."

page 6. It would be very helpful to give a short description of how a pi-A isotherm can be interpreted. I imagine most readers, like myself, are not familiar with reading these types of figures. What information does the isotherm reveal? What is a lift-off area? What is a collapse? How are these determined from the isotherm? Why are these quantities important? Amending Figure 1 to show a molecular picture of the various stages in the pi-A isotherm would help.

Response:

Langmuir films are formed when surfactants are spread at the air–water interface. Surfactants are amphiphilic molecules with hydrophobic tails and hydrophilic heads. Since their tails of surfactants are hydrophobic, their exposure to air is favored over that to water. Similarly, since the heads are hydrophilic, the head–water interaction is more favorable than air–water interaction. The overall effect of such packing is reduction in the surface energy. When surfactant concentration is less than the minimum surface concentration of collapse, the surfactant molecules can arrange a monolayer on the surface of water.

Surface pressure–area isotherms show phase transition-like behaviour of the Langmuir films. In the gas phase, there is minimal pressure increase for a decrease in area. The first transition from gas phase to liquid phase occurs at the lift-off area. There is a proportional increase in surface pressure with decreasing area. Moving into the solid region of monolayer is accompanied by another sharp transition to a more severe area dependent pressure. This trend continues up to a point where the molecules are packed closely and have very little space to move. Applying an increasing pressure at this point causes the monolayer to become unstable and destroy the monolayer. The surface pressure during the monolayer collapse may remain approximately constant (in a process near the equilibrium) or may decay abruptly (out of equilibrium - when the surface pressure was over-increased because lateral compression was too fast for monomolecular rearrangements).

We have added some description of surface pressure-area isotherm and the inserts in Figure 2 to show the phase behaviours on Page 8:

"The monolayers usually composed of amphiphilic molecules with a hydrophilic head and a hydrophobic tail are assembled vertically at the air–water interface. Surface pressure–area ($\pi$−A) isotherms show phase transition behaviours of organic films. Owing to the amphiphilic characteristics of phospholipids, the head groups of DOPC and DSPC molecules prefer to be in the solution while their tails stretch into the air. The $\pi$−A isotherms recorded for DOPC and DSPC monolayers on artificial seawater with and without photosensitizers are shown in **Fig. 2(a)** and **(b)**, respectively. When the surface area of the DOPC monolayer was larger than 125 Å$^2$/molecule, the distance between the DOPC molecules was quite large and the intermolecular force was quite weak. The surface pressure of a DOPC monolayer on artificial seawater started to increase from 125 Å$^2$/molecule, where the DOPC monolayer surface state underwent a transition from the gas to the liquid-condensed phase. After the first phase transition, there is a proportional increase in surface pressure with decreasing area. This caused condensation and ordering at the interface, increasing the surface pressure of the organic monolayer. This trend continued up to a point where the DOPC molecules were

packed closely and have very little space to move. Finally, the DOPC monolayer collapsed at 46 mN/m (Pereira et al., 2018). Applying an increasing pressure at the collapse pressure caused the monolayer to become unstable and destroy the monolayer. The packing and phase behaviours of DOPC monolayer at different pressures were also shown in the inserts in **Fig. 2(a)**."

page 9, line 12. Indicate that this is 30% after 90 min of irradiation.
Response:
We have modified the sentence on Page 11:
"The relative area of the DOPC monolayer on pure artificial seawater was reduced to 30% after 90 min irradiation."

lines 12 – 14: "There was evidently.." This statement is contrary to the data: the addition of photosensitizer appears to decrease the decay of the DOPC monolayer. This sentence is then contradicted by the next sentence ("The presence of…").
Response:
The decay rate of the DOPC monolayer in the artificial seawater containing photosensitizers was slower than that on the subphase of pure artificial seawater. The presence of photosensitizing molecules in the subphases decreased the loss of the molecular area for the DOPC monolayer. To avoid misunderstanding, we have modified the sentence on Page 11:
"It is evident that the presence of photosensitizing molecules in the subphases decreased the loss of the molecular area for the DOPC monolayer."

»Minor points page 1, lines 20-22: Define OA and EA. Also, the sentence is unclear. What is the comparison? line 24: Since there is no direct experimental evidence for hydroperoxidation in the current work, this statement should be qualified. line 28: "the processing of organic aerosol aging" does not "control" aerosol composition.
Response:
We have modified the statement on Page 1:
"The oleic acid (OA) monolayer possessing a *cis* double bond in an alkyl chain is more expanded than elaidic acid (EA) monolayers on artificial seawater that contain a photosensitizer."
"Instead, the photochemical reaction initiated by the excited photosensitizer and molecular oxygen can generate new unsaturated products in the DOPC monolayers, accompanied by an increase in the molecular area."
"The results of $PM_{2.5}$ and SOA samples will contribute to our understanding of the processing of organic aerosol aging that alters the aerosol composition."

page 2, l.28: This is poorly worded: the triplet state is not susceptible to oxidation by a hydrocarbon.
Response:
We have corrected the sentence on Page 2:

"Therefore, photosensitizers can contribute to organic aerosol aging and growth when generating a triplet excited state that can oxidize hydrocarbon upon absorbing light."

page 5, line 20: Is TUV the model number of the lights? If not, what is model number? What was the photon flux in the sample?
Response:
The trough was surrounded by three UV fluorescent lamps (UVA range, Philips TUV TL-Mini 8W, 31 cm length, 2.6 cm o.d.). The light spectrum of the UV fluorescent lamps ranged from 300 to 420 nm with peak intensity at 365 nm, which was similar to the irradiation of solar UV band. We have modified the sentence on Page 7:
"The trough was illuminated by three UV fluorescent lights (Philips TUV TL-Mini 8 W) with peak intensity at 365 nm. The lamps were located above the Langmuir trough at a distance of 20 mm."

Figure 1: The structures are very small and difficult to discern, especially the double bonds.
Response:
We have modified the structures of lipids in Figure 1.

page 6, line 17: This sentence is not precise enough: the entire DOPC molecule didn't go from gas phase to aqueous phase.
Response:
Gas phase and liquid phase are corresponding to the different phase behaviours of lipid monolayer. We have added some inserts in Figure 2 to show different phase behaviours of lipid monolayer on Page 9.

page 7, line 5: "in both the liquid and condensed phase". How is "condensed" phase different from "liquid" phase?
Response:
We have corrected the sentence and added the inserts in Figure 2 to show the different behaviours of DOPC monolayer on Page 9:
"The $\pi$–A isotherms of the DOPC monolayer on artificial seawater mixed with SOA sample and IC overlaps in the liquid-condensed phase."

line 14: "for the DPPC monolayer". Shouldn't this be DOPC?
Response:
We have corrected the sentence as suggested.

line 17: "The introduction of photosensitizers…had a profound effect…". This is only true for HA, not for IC.
Response:
We have modified the sentence on Page 9:
"The introduction of HA into the subphase also had a profound effect on the shape of $\pi$−A isotherms for DSPC monolayers."

---

## Author Comment (AC3) · 3 Jun 2019

We have revised our manuscript according to the suggestions of the Referee's comments and the responses to the comments are as following. For clarity, the Referee's comments are reproduced in blue, authors' responses are in black and changes in the manuscript are in red color text.

This study presents an investigation of the influence of photosensitizing species (both model species and ambient samples) on the properties of long-chain fatty acids at the surface of model seawater. Specifically, the authors couple Langmuir trough measurements of surface pressure and molecular area with PM-IRRAS compositional measurements to explore irradiation-induced changes in monolayer physical and chemical properties.

Although this is an interesting study, I have a number of major concerns regarding the manuscript in its current form that I believe should be addressed prior to publication:

First, insufficient information is provided regarding the quantity of photosensitizers employed, specifically in the cases of PM and SOA. It is therefore unclear whether differences in the four photosensitizer types employed arise as a result of differences in photosensitizer properties or simply differences in photosensitizer concentration.
Response:
We have determined the concentration of $PM_{2.5}$ and SOA sample. The processes in detail were described below. We agree that the photosensitizing efficiency also depends on the concentration of the photosensitizer, and have deleted the relevant discussion about the photosensitizing efficiency.

Second, in the absence of information regarding experiment reproducibility, it is difficult to assess whether the reported results are meaningful. For example, Figure4c does not provide error bars, and many of the spectral shifts reported are very small ($\sim$ 2 cm-1)—do these results reflect variability between experiments, or real effects?
Response:
The spectral resolution of 8 cm$^{-1}$ is commonly used in the IRRAS measurements (*Langmuir*, **1996**, 12, 1027-1034; *J. Phys. Chem. B*, **2004**, 108, 15238-15245; *J. Phys. Chem. B*, **2005**, 109, 7428-7434). The $v_{as}(CH_2)$ and $v_s(CH_2)$ frequencies are known to be sensitive to the conformation order of hydrocarbon chains (*Spectrochim. Acta A* **1978**, 34, 395-406). Sometimes the shift of wavenumber can be lower than the resolution. It has been reported that $v_{as}(CH_2)$ vibrational mode of DPPC monolayer on $Zn^{2+}$ had a significant shift (3 cm$^{-1}$) to lower wavenumbers, and $Sr^{2+}$ had a $\leq$1 cm$^{-1}$ shift for the DPPC monolayer in the liquid-expanded phase. In the liquid-condensed phase (40 mN m$^{-1}$), peak shifts relative to DPPC on water were small ($\leq$1 cm$^{-1}$) (*Phys. Chem. Chem. Phys.* **2016**, 18, 32345-32357). The spectra were collected at a resolution of 4 cm$^{-1}$, however, all the shifts they discussed were smaller than 4 cm$^{-1}$. It indicates that the position of $v_{as}(CH_2)$ is not significantly different in the presence of these ions. Other

peak shifts relative to pure water are more significant. Spectra were averaged over 2000 scans, and IRRAS measurements were repeated at least three times to ensure reproducibility. We also highlight the appearance of new bands round 3000 cm$^{-1}$ after 90 min irradiation.

To a certain extent, the measurements will be influenced by environmental factors, such as temperature and relative humidity. Air conditioner was applied to ensure that the room temperature changed little during the measurement. The rate of the degradation may vary from day to day, but the general trends were the same. The data presented in the manuscript were recorded during the same day and are a representative selection of many repeats. At least three independent runs were performed to check the reproducibility, especially the lifting area, the phase transition point and the collapse pressure. The trough was cleaned by repeatedly rinsing with ethanol and Millipore water before use. We have given an example of π-A isotherms of DOPC monolayers on pure artificial seawater in **Fig. S2** in the supplement. Standard deviations of the molecular area and surface pressure were ±1 Å$^2$/molecule and ±2 mN/m, respectively. To avoid the change of surface area among each independent measurement of surface pressure-area isotherm, we used relative area to compare. We ensured that the change of relative area was due to the irradiation.

Third, the manuscript often mixes results and discussion together, and is therefore difficult to follow. I provide specific examples of this in my comments below.
Response:
We have rewritten the results and discussion as suggested.

Fourth, the conclusions/mechanisms in the latter half of the paper are overstated in the context of the results provided. Specifically, I believe that insufficient experimental evidence is provided for the production of hydroperoxides.
Response:
The mechanism of photosensitizing reaction of unsaturated lipids were proposed based on previous studies (*Langmuir*, **2007**, 23, 1307-1314; *Biophys. J.*, **2009**, 97, 1362-1370; *Colloid Surface B*, **2018**, 171, 682-689). Hydroperoxide is generally regarded as the primary product of unsaturated lipid. However, the PM-IRRAS technique was not sufficiently sensitive to detect the hydroperoxide bands directly. The new bands of HC=CH group appearred after irradiation are assigned to the DOPC hydroperoxide. DOPC hydroperoxides with OOH groups were more hydrophilic than DOPC. The OOH groups can move into aqueous solution, which resulted in the increase of molecular area. The increase in relative area of DOPC monolayer in the artificial seawater containing photosensitizers under irradiation also support the mechanism. We believe that the results are consistent with (and not proof of) hydroperoxidation.

Besides, we have tried to detect the photochemical products of DOPC by GC-MS. The Agilent Technologies 5977B mass selective detector equipped with an electron ionization (EI) source was used to perform MS measurement. The ionization energy and the temperature of the EI source were 70 eV and 230 °C, respectively. Sample solution with volume of 4 μL was injected into the gas chromatograph equiped with

an HP-5MS capillary column (30 m × 0.25 mm id × 0.5 mm film thickness; Agilent Technologies, Santa Clara, CA). The injector temperature was set as 300 ℃ to allow the sample solution to evaporate completely. The time delay of the solvent was 2.5 min to prevent solvent from getting into the MS detector. The GC oven temperature was programmed as follows: the temperature was maintained at 80 ℃ for 1 min, then heated at a rate of 5 ℃/min$^{-1}$ to 240 ℃ and maintained for 1 min. Sample were analyzed in full scan mode with mass (m/z) range from 50 to 500. The mass peak at m/z=281 can be explained as oleic acid tail in DOPC. The weak mass peak at m/z=313 correspond to benefit of 32 Da, which can be explained as OO. However, due to the limitation of the instruments, hydroperoxides were not directly identified.

In the following sections, I outline additional specific scientific questions/issues regarding the manuscript, organized by manuscript section.

**Abstract**
P1L1 Photosensitizing compounds don't *contain* brown carbon; this is confusing as written.
Response:
We have modified the sentence on Page 1:
"Photosensitizing compounds like brown carbon can absorb UV light and produce low volatile organic compounds (O:C ratio of 0.25 to 1) at the surface of aqueous particles."

P1L1 How is "low volatile" defined here? In addition, citing only energy transfer here seems unnecessarily specific.
Response:
We have modified the statement on Page 1:
"Photosensitizing compounds like brown carbon can absorb UV light and produce low volatile organic compounds (O:C ratio of 0.25 to 1) at the surface of aqueous particles."

P1L21 OA/EA need to be defined prior to use (*i.e.* in L17).
Response:
We have modified the statement on Page 1:
"The oleic acid (OA) monolayer possessing a *cis* double bond in an alkyl chain is more expanded than elaidic acid (EA) monolayers on artificial seawater that contain a photosensitizer."

P1L24  I think that the experimental evidence for hydroperoxidation is weak; the abstract would be stronger, in my opinion, if it focused more closely on the results.
Response:
We have modified the statement on Page 1:
"Instead, the photochemical reaction initiated by the excited photosensitizer and molecular oxygen can generate new unsaturated products in the DOPC monolayers, accompanied by an increase in the molecular area."

P1L29   This final sentence is too vague, I think.

Response:

We have modified the sentence on Page 1:

"The results of $PM_{2.5}$ and SOA samples will contribute to our understanding of the processing of organic aerosol aging that alters the aerosol composition."

**Introduction**

P2L4 Are these the only two fates available to phospholipids?

Response:

As an organic compound in the ocean, phospholipids can participate in the life cycle through digestion of organisms, and atmospheric circle through transfer into the atmosphere. These two fates of phospholipids we mentioned are relevant to the atmosphere. We have modified the sentence on Page 2:

"Phospholipids can be transported from seawater into sea spray aerosol (SSA) directly. They also can be further transformed into fatty acids through heterotrophic breakdown."

P2L6 Perhaps some more recent references would be helpful here?

Response:

We have added some more recent references on Page 2:

"Long chain saturated fatty acids, such as palmitic acid and stearic acid, correspond to major constituents of the sea surface microlayer and are also detected in marine aerosols (Marty et al., 1979; Slowey et al., 1962; Wu et al., 2015; Hu et al., 2018; Kang et al., 2017)."

P2L7 This is misleading as written—I think that it should clarify *which* component of seawater unsaturated fatty acids dominate. As is, it makes it sound as though unsaturated fatty acids are a major fraction of seawater.

Response:

We have modified the sentence on Page 2:

"Unsaturated fatty acids with carbon chain lengths of 18-22 carbons also dominate the organic composition of seawater samples (Jeffrey, 1966; Osterroht, 1993)."

P2L11 Given that the focus of the paper is reactions at the sea surface, I think that the cooking-related references are unnecessary.

Response:

We have removed the discussion about the cooking-related references.

P2L17 Is all WSOC necessarily BrC? I think that there is also non-absorbing WSOC?

Response:

WSOC we mentioned in the introduction is the water-soluble component of brown carbon. We have modified the statement on Page 2:

"The fully dissolved organic fraction of BrC is referred to as light-absorbing water-soluble organic carbon (WSOC), while colloidal aggregates belong to water-insoluble organic carbon (WIOC)."

P2L20–22 I wonder if it might be useful to outline BrC sources that do not involve ammonia/ammonium.

Response:

We have added some discussion about other BrC without ammonia/ammonium on Page 2:

"The products from aqueous OH oxidation of phenolic compounds have characteristic BrC absorption spectra (Gelencser et al., 2003; Vione et al., 2014). The SOA generated by high-NOx photooxidation and ozonolysis of monoterpenes have absorption in the wavelength range of 355−780 nm (Nakayama et al., 2013; Flores et al., 2014)."

P2L26 I don't think that it's correct to say that photosensitizers are a *subset* of BrC, as there are photosensitizers that do not fall under the BrC umbrella.

Response:

We have modified the statement on Page 2:

"Photosensitizers in the atmosphere can absorb and convert the energy of photons into chemical energy that can facilitate reactions at aerosol surfaces (George et al., 2015)."

P2L32 I would rephrase "just a few radical reactions"—I'm not exactly sure what is being argued here.

Response:

We have modified the statement on Page 2-3:

"Traces of photosensitizing species in the aerosol phase, such as imidazoles, quinones and nitrophenols also contribute to SOA formation through their promoting effect with radical reactions (Li et al., 2016; Desyaterik et al., 2013; Pillar and Guzman, 2017)."

P3L1 I think that it might be clearer to say that IC is a *component* of BrC?

Response:

We have modified the statement on Page 3:

"Imidazole-2-carboxaldehyde (IC) is a component of BrC (Ackendorf et al., 2017; Arroyo et al., 2018; Rossignol et al., 2014), produced through the aqueous reaction of glyoxal or methylglyoxal with ammonium sulphate (Aiona et al., 2017)."

P3L5–20 A transition between IC and HULIS is missing here. In addition, some detail regarding HULIS seems unnecessary here (*e.g.* the portions relating to combustion emissions).

Response:

We have deleted the description about the emission of HULIS and modified the statement on Page 3:

"These water-soluble organic materials like IC are termed humic-like substances (HULIS) due to their similar properties to macromolecular humic substances, such as their amphiphilic and polyacidic nature, aromaticity, surface active properties and light absorption ability (Gelencser et al., 2002; Graber and Rudich, 2006; Sannigrahi et al., 2006; Krivacsy et al., 2008). HULIS correspond to 10−35% of fine organic materials

in atmospheric aerosols and account for up to 72% of WSOC in some ambient aerosol samples (Emmenegger et al., 2007; Feczko et al., 2007). Humic substances (HS) consist of three operationally defined components: humic acids (HA), fulvic acids, and humins. These HS represent a fraction of the molecularly uncharacterized component of dissolved organic matter in the ocean (Zhu et al., 2017b). HA in the ocean are widely believed to derive primarily from the products of marine phytoplankton degradation and less so from terrestrial sources (McCarthy et al., 1996). For primary marine aerosols, WSOC containing HULIS components was suggested to originate from bubble-bursting at the surface of seawater, which transfers organic matter into marine aerosol particles (Cavalli et al., 2004; Yu et al., 2004). HULIS exist primarily in the droplet mode with aerodynamic diameter in the range of 0.7-0.8 μm (Wang and Yu, 2017)."

P3L32 "processing the aging"—should be reworded.
Response:
We have reworded the sentence on Page 4:
"The stability behaviour of these organic films under irradiation and the impact of photosensitizers on organic aerosol aging are presented and discussed."

**Experimental**
P4L10 What does the 90% purity level for humic acid refer to? How was it determined? Is structural information available for this humic acid standard (*i.e.* how does it compare to other HA sources, like Suwanee River humic acid / Aldrich humic acid)?
Response:
Humic acid sample purchased from Aladdin contains > 90% fulvic acid. Its number-average molecular weight is 1032. Suwanee River humic acid and Aldrich humic acid are unavailable for us. The light absorption of humic acid have been measured by UV–vis spectrophotometer. We have modified the statement on Page 4:
"The photosensitizers imidazole-2-carboxaldehyde (IC) (97%, Alfa Aesar) or HA (> 90% fulvic acid, Aladdin) were added into the ASW with concentrations of 2.5 mM and 30 mg/L, respectively."

P4L12 How were the aqueous loadings of PM2.5/SOA determined? How were the reported concentrations of IC (2.5 mM) and HA (30 mg/L) chosen? Could differences between the various samples reflect different *amounts* of additives rather than differences in the properties of the various additives? Some additional information would be useful here in this context.
Response:
Separation of the water-soluble and insoluble fractions in $PM_{2.5}$ sample was performed following the method described elsewhere (*Sci. Total Environ.*, **2016**, 542, 36–43580). We collected SOA sample onto aluminum foils by DLPI. The aluminum foils were sonicated by placing a reaction flask with the ultrapure water (40 mL) in an ultrasonic bath by 15 min. Then, the upper extract solution was concentrated to dry by rotary evaporator and termovap sample concentrator. The dried residue was re-dissolved with artificial seawater.

The concentration of 30 mg/L humic acid is commonly used in the photochemical study (*Environ. Sci. Technol.*, **2015**, 49, 13199-13205; *Sci. Rep.*, **2015**, 5, 12741). In the previous studies, larger than or equal to 30 mg/L humic acid were added directly into the artificial seawater to mimic the presence of the dissolved organic matter in the sea surface microlayer (*Sci. Rep.*, **2015**, 5, 12741; Chiu, R., *Geophys. Res. Lett.*, **2017**, 44, 1079–1087). In this investigation, the concentration of humic acid in fresh sea spray is considered close to the real seawater.

These dissolved organic matters are present in different and varying concentrations in seawater and aerosols. The concentrations of IC commonly used in the experiment varied from 0.25 mM to 0.6 M (*Environ. Sci. Technol.*, **2018**, 52, 7680−7688; *Faraday Discuss.*, **2013**, 165, 123–134; *C. R. Chimie*, **2014**, 17, 801–807).

We have added some explanation on Page 5-6:

"All the filters were dissolved into 40 mL ultrapure water with ultrasonic agitation. Sonication was performed in an ultrasonic bath with a frequency of 40 kHz and the power of 80 W. The sonication time was 15 min. Subsequently, the suspension was centrifuged at 1780 g for 40 min. The supernatant, which contains the water-soluble fraction including water-soluble organic compounds (WSOC) and inorganic ions, was re-collected by freeze-drying. The insoluble fraction separated from soluble fraction was also freeze-dried. The mass ratio of insoluble to soluble fraction is 0.91:1. Then, 3.3 mg of freeze-dried soluble sample was dissolved in 1000 mL artificial seawater. The concentration of $PM_{2.5}$ sample in the artificial seawater is 3.3 mg/L."

"Then, the SOA samples collected on the aluminium foil pieces were dissolved in ultrapure water by sonicating for 1 min in an ultrasonic bath. The extract water solution was concentrated by rotary evaporation. The residue was dried under high purity nitrogen stream. Then, the SOA sample was transferred to the artificial seawater with the concentration of 0.66 mg/L."

P5L4 What were the NO concentrations employed? Which type of $NO_x$/VOC environment were the authors attempting to reproduce here, and why?

Response:

We detected the NO concentration by $NO$-$NO_2$-$NO_x$ analyzer (Model 42C, Thermo Electron Corporation, USA). The initial concentration of $NO_x$ and limonene are 206 ppb and 684 ppb, respectively. The initial NO concentration is 164 ppb. The SOA formation of limonene photooxidation experiments was performed under high-$NO_x$ conditions which was defined as $[NO_x]_0 > 30$ ppb, $[VOC]_0/[NO_x]_0 < 10$) (*Atmos. Chem. Phys.*, **2016**, 16, 11237–11248). The effects of $NO_x$ on the formation of BrC SOA during toluene photo-oxidation were emphasized (*Atmos. Chem. Phys.*, **2007**, 7, 3909–3922). BrC chromophores in SOA generated from photochemical chamber was enhanced under high $NO_x$ condition (*Phys. Chem. Chem. Phys.*, **2015**, 17, 23312–23325).

We have modified the statement on Page 5:

"The SOA formation of limonene photooxidation experiments was performed under high-$NO_x$ condition (Sarrafzadeh et al, 2016). The initial concentration of $NO_x$ and NO detected by $NO$-$NO_2$-$NO_x$ analyzer (Model 42C, Thermo Electron Corporation, USA)

are 206 ppb and 164 ppb, respectively."

P5L7 Would one expect different compositions for the different size fractions? Were all size fractions employed? Would one anticipate any SOA compositional biases induced by collecting under low-pressure conditions?
Response:
The diameters of particles collected by 13-stage Dekati low-pressure impactor (DLPI) range from 0.016 μm to 10 μm. The compositions for the different size fractions may be different. The collection under low-pressure conditions did not induce SOA compositional biases. All size fractions of SOA we collected were involved in photosensitizing reaction.

P5L12   Why was chloroform employed as solvent?
Response:
Chloroform is a common solvent used in the Langmuir experiments. It can dissolve most of organic compounds. When chloroform solution was dropped onto the surface of water, chloroform can evaporate quickly.

P5L15   A reference for the Wilhelmy plate method, as well as details of the method, would be useful.
Response:
Wilhelmy plate method is the most common way to determine the surface/interfacial tension and widely used in the preparation and monitoring of Langmuir–Blodgett films (*J. Colloid Interf. Sci.*, **1997**, 185, 245–251; *J. Chem. Eng. Data*, **2006**, 51, 255-260; *Atmos. Environ.*, **2010**, 44, 329-337). A Wilhelmy plate is a thin, generally rectangular plate made of iridium–platinum or filter paper with a few centimeters in length and height. The plate is attached to a force sensor on the one side of barrier. In the measurement procedure, the Wilhelmy plate is first dipped into the liquid and then pulled back to the position of first contact. In this method, the plate is oriented perpendicular to the interface, and the force exerted on it is measured. Adding a monolayer to the surface reduces the surface tension, and the surface pressure, $\pi$ is given by the following equation:

$$\pi = \gamma_0 - \gamma$$

where $\gamma_0$ is equal to the surface tension of the water and $\gamma$ is the surface tension due to the monolayer.
We have added some details of Wilhelmy plate method on Page 6-7:
"A Wilhelmy plate is a rectangular plate made of filter paper with a few centimeters in length and height. The plate is attached to a force sensor on the one side of barrier. In the measurement procedure, the Wilhelmy plate is first dipped into the liquid and then pulled back to the position of first contact to measure the surface pressure (Rame, 1997; Hyvärinen et al., 2006; Aumann et al., 2010). The surface pressure was obtained based on the following equation:

$$\pi = \gamma_0 - \gamma, \tag{3}$$

where $\gamma_0$ is equal to the surface tension of the solution and $\gamma$ is the surface tension due

to the monolayer."

P5L18 How were the uncertainties in surface pressures (+/- 2 mN/m) estimated?
Response:
Organic compounds usually have more positive responses than inorganic materials for outside factors, such as pressure, temperature or gas change. Surface pressure-area isotherms were also influenced by the environmental condition, such as temperature and relative humidity. Our experiments are processed in a sealed glass cover. We repeated three times of surface pressure-area measurements for the DOPC monolayer. Standard deviations of the molecular area and surface pressure were $\pm 1$ Å$^2$/molecule and $\pm 2$ mN/m, respectively.

P5L20 How did the illumination intensity compare to ambient conditions? How far away from the Langmuir troughs were the lamps?
Response:
The light spectrum of the UV fluorescent lamps ranged from 300 to 420 nm with peak intensity at 365 nm, which was similar to the irradiation of solar UV band. The lamps were located 20 mm above the surface of solution.
We have added some description on Page 7:
"The trough was illuminated by three UV fluorescent lights (Philips TUV TL-Mini 8 W) with peak intensity at 365 nm. The lamps were located above the Langmuir trough at a distance of 20 mm."

P5L26 Why was this incident angle (40 degrees) chosen for the light beam? It is my understanding that PM-IRRAS is often performed at a glancing angle? In addition, I think that some more information regarding the PM-IRRAS technique would be useful here (*i.e.* which measurements provide the PM-IRRAS signal).
Response:
The intensity of IR spectra can change with the incident angle of IR beams. Based on our previous investigation, we found that the spectra obtained at the incident angle of 40 degree have intensive peak and smooth baseline (*Sci. Total Environ.*, **2017**, 580, 1155-1161). Therefore, we chose the incident angle of 40° for the light beam in the follow-up investigations. We have added some introduction about IRRAS on Page 7:
"IRRAS spectra of organic film are generally presented as plots of reflectance-absorbance against wavenumber. Reflectance-absorbance is defined as $-\log_{10}(R/R_o)$ where R is the reflectivity of the film-covered surface and $R_o$ is the reflectivity of the aqueous subphase. The incident beam is directed onto the aqueous solution surface in the Langmuir trough at a 40° angle based on our previous study (Li et al, 2017). Then, the reflected beam is measured by the MCT detector."

**Results and discussion**
P6L10 How is the light absorption relevant to the discussion of packing/phase behaviour?
Response:

The surface pressure-area isotherms were measured without irradiation. These two sentences may not proper in the section of packing and phase behavior. We have moved them to experimental section on Page 7.

P6L12–P7L6    This section mixes results and discussion of results in a way that I find confusing. I think that it would be clearer for the reader if the isotherms were first described, and then the implications of the isotherm shapes were explained.
Response:
We have rewritten this section and added some discussion about isotherm shapes as suggested on Page 8:
"The monolayers usually composed of amphiphilic molecules with a hydrophilic head and a hydrophobic tail are assembled vertically at the air–water interface. Surface pressure–area ($\pi$−A) isotherms show phase transition behaviours of organic films. Owing to the amphiphilic characteristics of phospholipids, the head groups of DOPC and DSPC molecules prefer to be in the solution while their tails stretch into the air. The $\pi$−A isotherms recorded for DOPC and DSPC monolayers on artificial seawater with and without photosensitizers are shown in **Fig. 2(a)** and **(b)**, respectively. When the surface area of the DOPC monolayer was larger than 125 Å$^2$/molecule, the distance between the DOPC molecules was quite large and the intermolecular force was quite weak. The surface pressure of a DOPC monolayer on artificial seawater started to increase from 125 Å$^2$/molecule, where the DOPC monolayer surface state underwent a transition from the gas to the liquid-condensed phase. After the first phase transition, there is a proportional increase in surface pressure with decreasing area. This caused condensation and ordering at the interface, increasing the surface pressure of the organic monolayer. This trend continued up to a point where the DOPC molecules were packed closely and have very little space to move. Finally, the DOPC monolayer collapsed at 46 mN/m (Pereira et al., 2018). Applying an increasing pressure at the collapse pressure caused the monolayer to become unstable and destroy the monolayer. The packing and phase behaviours of DOPC monolayer at different pressures were also shown in the inserts in **Fig. 2(a)**."

Were multiple trials performed? What are the uncertainties associated with the (rather precise) values reported here (*e.g.* 46 mN/m for DOPC monolayer collapse)?
Response:
As mentioned above, we have repeated at least three times to check the reproducibility, especially the lifting area, the phase transition point and the collapse pressure.

How is it meaningful to compare collapse pressures for HA vs. IC when the two species were added in different quantities?
Response:
We compared the collapse pressures of DOPC monolayer on the artificial seawater containing photosensitizers to that on pure artificial seawater. The collapse pressures for HA and IC were lower than that for pure artificial seawater. We have modified the statement on Page 9:

"The collapse pressure of DOPC monolayer on pure artificial seawater decreased from 46 to 28 mN/m with the addition of IC. The collapse pressure for the DOPC monolayer on the artificial seawater containing HA was even lower."

Why were the collapse pressures higher for PM2.5/SOA vs. IC/HA? I think that some mechanistic discussion of these differences is warranted—what is the take-home message for the reader, here?
Response:
In addition to water soluble organic compound, $PM_{2.5}$ and SOA samples also contain inorganic salts. The presence of inorganic ions in the $PM_{2.5}$ and SOA samples may contribute to the assembly of organic monolayer. We have added some discussion about the difference of collapse pressures on Page 9:
"In contrast to IC and HA, the collapse pressures of DOPC monolayer were higher in the $PM_{2.5}$ and SOA samples. The inorganic ions from the $PM_{2.5}$ and SOA samples may contribute to the assembly of organic monolayer."

What does "coincided" mean, on P7L4? Is this a meaningful observation?
Response:
There has a small overlap of the π–A isotherms between the DOPC monolayer on the artificial seawater containing SOA and IC samples. We have modified the statement on Page 9:
"The π–A isotherms of the DOPC monolayer on artificial seawater mixed with SOA sample and IC overlaps in the liquid-condensed phase."

P7L12 How is this collapse pressure (55 mN/m) determined, exactly? To me, it is not clear that the surface pressure is decreasing (the decrease could perhaps be noise?)
Response:
The rapid decrease of surface pressure for organic monolayer can be considered as the collapse pressure of organic film. According to Figure 2, it seemed that after it collapsed, the surface pressure of DSPC monolayer started to increase again a few seconds later. The secondary increase of surface pressure indicated that it tends to form multiple layer of organic film. We have added some explanation on Page 9:
"The appearance of the secondary increase in surface pressure after DSPC monolayer collapsed, indicated that DSPC monolayer had a tendency to form multiple layer."

P7L14   DPPC should read DOPC, I think.
Response:
We have corrected it on Page 9:
"Although the structures of DSPC and DOPC are quite similar, the lift-off area for the DSPC monolayer on artificial seawater was smaller than that for the DOPC monolayer in **Fig. 2(b)**."

P7L20   Again, is there an uncertainty associated with this lift-off area?
Response:

As mentioned above, the standard deviation of the lift-off area was $\pm 1$ Å$^2$/molecule.

Response:

The lift-off area can be defined as the area when the surface pressure of organic monolayer starts to increase. When the area reduced to the molecular area of 82 Å$^2$/molecule, both the surface pressure of EA and OA monolayers began to increase. We have added the definition of lift-off area on Page 9:

"The lift-off area is the area when the surface pressure of organic monolayer starts to increase."

Response:

The surface pressure curve of OA monolayer on the pure water increases steadily towards a plateau-like region (*J. Geophys. Res.-Atmos.*, **2007**, 112). The OA monolayer on the artificial seawater also proceeded through a steadily increased but not plateau-like region at the surface pressure of approximately 20 mN/m, as can be seen in Fig. 3. Therefore, the curve of OA monolayer has no obvious collapse in this experiment. To be more accurate, we have modified the statement on Page 10:

"However, the EA monolayer can reach a maximum surface pressure of 34 mN/m, which is higher than that of OA (below 20 mN/m)."

Response:

We have rewritten this paragraph as suggested on Page 10-11:

"In the presence of IC in the artificial seawater, the fatty acid monolayers shifted to smaller areas and had higher collapse pressures relative to pure artificial seawater. The lift-off area value of 60 Å$^2$/molecule for EA monolayer on artificial seawater containing IC is larger compared to saturated SA monolayer on the same subphase (28 Å$^2$/molecule). The EA monolayers on the artificial seawater containing IC transitioned from TC to UC phase at the surface pressure of about 7-8 mN/m during the monolayer formation. Additionally, the EA monolayer exhibits the TC and UC phase transition below 17 °C (Iimura et al., 2001). The EA monolayer on the artificial seawater containing IC collapsed at the surface pressure of 46 mN/m. The collapse of the OA monolayer on the artificial seawater containing IC occurred in the TC state at a surface pressure of 24 mN/m. The behaviour in the UC phase of the EA monolayer is similar to that of the SA monolayer, reflecting that the photosensitizer molecules may be squeezed out of the tail of EA at high surface pressure. There is a remarkable obstructive

effect on the ordering of organic film for the film-forming molecule that contain a *cis*-double bond. The *cis*-double bond in OA disturbs close packing of the molecules and weakens the chain-chain attractive interaction (Iimura et al., 2001). Therefore, OA has a greater tendency to form expanded monolayers."

P9L2 It would be useful for the non-expert reader to explain (perhaps in the methods) what actually *happens* in these experiments, i.e. how does one measure an area relaxation curve, experimentally?
Response:
We set 25 mN/m as target pressure and observed the change of surface area with time. To avoid the slight difference of surface area between two independent experiments, we compared the relative area change ($A/A_0$). $A_0$ represents the initial area for DOPC monolayer at the surface pressure of 25 mN/m. We have added some discussion about area relaxation curve in the experimental section on Page 7:
"For area relaxation measurements, the changes of relative molecular area ($A/A_0$) with time were also recorded, where $A$ is the measured molecular area and $A_0$ is the initial area for DOPC monolayer when the compression to 25 mN/m was reached."

P9L9 Is this a reasonable timescale for lipid oxidation? It seems quite fast to me?
Response:
To monitor the degradation of the phospholipids, the time-dependent change of the molecular area at constant surface pressure were measured. This instability may be attributed to spontaneous degradation by oxidation mediated by various reactive species in the air. At a relative area of approximately 18%, the barriers could not move anymore and the experiment was terminated. The rate of the degradation varied from day to day, but the general trends were the same. In previous study, the relative area of DOPC monolayer was reduced to 55% after 20 min irradiation (*Langmuir*, **2016**, 32, 3766−3773). It was faster than our work.

P9L12 Where is this 30% value taken from? Is it at 90 min?
Response:
The relative area of the DOPC monolayer on pure artificial seawater was reduced to 30% under 90 min irradiation. We have modified the sentence on Page 11:
"The relative area of the DOPC monolayer on pure artificial seawater was reduced to 30% after 90 min irradiation."

P9L12 What does a "much greater change" in the decay rate mean? It would be helpful to be more explicit here about the *direction* of this change.
Response:
We have modified the sentence on Page 11:
"It is evident that the presence of photosensitizing molecules in the subphases decreased the loss of the molecular area for the DOPC monolayer."

P9 Figure 4 I find Figure 4c quite unclear. What does the difference between the curves

mean, exactly? Why was this metric chosen here? I think that the division of data between the main text and the supplementary makes this section more confusing than it needs to be. In addition, how do the differences between samples compare to sample variability between trials?

Response:

We have updated Figure 4 in the revised manuscript.

P10L1–21 I think that this section is missing some interpretation of the results—why do these added species result in area increases for the monolayers, exactly? I think that these paragraphs would benefit from a clearer mechanistic explanation for the observed results—especially because, as I mention in my previous comment, the data are not intuitively displayed (*i.e.* much data is in the supplementary section); in the current form, I can't quite grasp the *meaning* of the results.

Response:

We have changed Figure 4 and modified these two paragraphs on Page 12-13:

"In the presence of photosensitizers, the relative areas of the irradiated DOPC monolayers became larger than those of the non-irradiated DOPC as summarized in **Fig. 4(f)**. It is clearly shown where the effect of the photosensitizer on both the DOPC monolayer in the dark and irradiation is illustrated as a relative increase in area ($\Delta$). The presence of IC and HA in the subphase yielded a relative area increase ($\Delta$) of 51% and 50% for DOPC monolayer, respectively. The similar increase of relative area was also observed in the presence of SOA or $PM_{2.5}$ sample. The relative increases of molecular area for the DOPC monolayer mixed with SOA sample were 43%. The relative area increase for the DOPC monolayer with the $PM_{2.5}$ sample after 90 minutes of irradiation reached 41%. The area loss observed for lipid monolayers for the different compositions of the subphases, at a constant surface pressure, is indicative of their stability (Avila et al., 1999). Compared to the experiments without irradiation, DOPC monolayers were considerably more stable upon irradiation with a photosensitizer. The results of relative area suggested that photosensitizers induce possible reactions of DOPC under irradiation.

There were no significant changes of the molecular areas for the DSPC monolayers with and without exposure to light (**Fig. S3-5**). In the irradiation experiments, the decrease of the DSPC monolayer area in artificial seawater with IC and HA is less than 5% after 90 minutes. The loss of the DSPC monolayer area after 90 minutes of irradiation is at least 13%, with respect to the subphase of pure artificial seawater without photosensitizers. In irradiation experiments, the relative areas of the DSPC monolayer on the artificial seawater mixed with IC or HA were much closer to those experiments without irradiation. Therefore, the stability of the DSPC monolayer on the artificial seawater containing photosensitizers did not change much between the irradiated and dark experiments. DSPC monolayers are typically more stable than DOPC ones, irrespective of irradiation. The smaller loss of molecular area suggested that the presence of photosensitizer molecules in the subphase improved the stability of the lipid monolayer relative to pure artificial seawater. The different results of relative area curve between the DOPC and DSPC monolayers implied that different reactions

were induced by photosensitizers under irradiation."

P10L30 Previously (P9L9), it was argued that the lipids underwent oxidation over the timescales of the experiments. If so, wouldn't this oxidation be visible using PMIRRAS?
Response:
As can be seen from Figure 5, the peak intensities of DOPC monolayer in pure artificial seawater (grey lines) with or without irradiation were weaker than others.

P10L33 How meaningful are these changes (*e.g.* 2923–2921 cm$^{-1}$)? Would changes in "chain ordering" be expected to lead to shifts to lower wavenumbers, and why?
Response:
These vibrational frequencies are well-known to decrease with the introduction of conformational order into the acyl chains. Thus, the decrease in the asymmetric CH$_2$ stretching frequency from 2923 to 2921 cm$^{-1}$ provides the information about the decrease of *gauche* rotamer in DOPC monolayer. The DOPC monolayer with more *trans* rotamer can be packed more ordered. The DOPC monolayer became more stable with lower energy, and subsequently the spectra were shifted to lower wavenumber.

P11L5 How meaningful is this change in peak height intensity ratio (1.61 to 1.82)? Were these peak height ratio changes observed for IC/HA? If not, why not? What sorts of mechanisms would be expected for each of these photosensitizers?
Response:
We have added some discussion on Page 13:
"In the case of IC in the subphase, the peak height intensity ratio between $I_{as}$ and $I_s$ in the DOPC monolayer increased from 1.61 to 1.82 due to irradiation. The increase of peak height ratio also occurred in the presence of HA. It indicated the order of the monolayer chains was increased."

P11L7 Again, how meaningful is a change from 3023 to 3020 cm$^{-1}$?
Response:
The shift from 3023 to 3020 cm$^{-1}$ indicates that DOPC monolayer packing was more ordered under irradiation. We have added some discussion on Page 13:
"The band of $v$(HC=CH) at 3023 cm$^{-1}$ was shifted to 3020 cm$^{-1}$ under the irradiation of the DOPC monolayer mixed with IC. It indicated that the aliphatic chains became more ordered under irradiation."

P11L12 How is the shift to lower wavenumbers related to the conformation order of the aliphatic chains? Some explanation would be helpful here.
Response:
We have added some explanation on Page 13:
"The shifts in CH$_2$ and CH$_3$ bands to lower wavenumbers indicate that the *gauche* rotamers in DOPC monolayer were decreased after 90 minutes of irradiation. Therefore, the conformation order of the aliphatic chains in DOPC monolayer was increased."

P12L9–P13L6 This section, in many places, says that spectral shifts are occurring due to "interactions/modifications", but doesn't clearly outline how these interactions would lead to the observed shifts. How does a shift from 967 to 942 cm$^{-1}$, for example, indicate modifications in DOPC monolayer packing?

Response:

These shifts of phosphate bands may be due to either hydrogen bonding of the phosphate group or water-induced structural rearrangements of the entire phospholipid headgroup (*J. Am. Chem. Soc.*, **1976**, 98, 851-853). In this work, these shifts in phosphate bands indicate that the interaction between photosensitizer molecules and DOPC molecules appears to have affected hydrogen bonding with water molecules in the neighborhood. The shift from 967 to 942 cm$^{-1}$ also indicated that photosensitizer molecules affected the hydration of DOPC head groups. We have added some discussion on Page 15:

"These shifts in phosphate bands indicated that the interaction between photosensitizer and DOPC molecules induced the dehydration of phosphate groups (Arrondo et al, 1984). In the presence of the SOA sample in the subphase, the antisymmetric stretching of the choline group $v_{as}(CN^+(CH_3)_3)$ band of irradiated DOPC monolayer was shifted from 967 to 942 cm$^{-1}$, which indicated that photosensitizer molecules affected the hydration of DOPC head groups."

P13L7 Why do the authors think that spectral shifts for DSPC were less significant than for DOPC?

Response:

We did not mean that the spectral shifts for DSPC were less significant. Overall, these spectral shifts induced by photosensitizers were more obvious for DOPC than for DSPC. The difference between DSPC and DOPC is that no new bands of DSPC monolayer after irradiation were observed. It indicated that DSPC films were less affected by irradiation. We have modified the sentence on Page 15:

"According to the comparison with the IRRAS spectra of DSPC monolayer in dark condition, no new bands of products were observed from the irradiated DSPC film. This result suggested that the DSPC films were less affected by irradiation."

P14–15 In my view, **Section 3.4** is not sufficiently grounded in the experimental results. How, exactly, do the results obtained support this rather complex proposed mechanism?

Response:

The new bands observed by IRRAS suggested that some new unsaturated products existed at the interface after irradiation. Additionally, other bands were changed little. We inferred that the photochemical product had a structure that is similar to DOPC. Based on the previous studies, unsaturated lipids are more susceptible to the attack of $^1O_2$ and free radicals. Reactions of $^1O_2$ with unsaturated bonds in lipid chains can generate hydroperoxide (OOH) groups (*Langmuir*, **2007**, 23, 1307-1314; *Biophys. J.*, **2009**, 97, 1362-1370; *Langmuir*, **2016**, 32, 3766-3773). This mechanism proposed here was generally accepted (*Biochim. Biophys. Acta.*, 1986, 857, 238-250; *Compr. Rev. Food. Sci. Food Saf.*, **2006**, 5, 169-186; *Compr. Rev. Food. Sci. Food Saf.*, **2017**, 16,

1206-1218; *Colloid Surface B*, **2018**, 171, 682-689). According to the mechanism, we tentatively assigned the unsaturated products measured by IRRAS as DOPC hydroperoxide. We have added some discussion on Page 16:

"The new band corresponding to CH stretching vibration of HC=CH group was measured by IRRAS, while other bands were changed little in **Fig. 5**. We infer that the structure of photochemical products is similar to DOPC. According to the mechanism of photosensitizing reaction, the unsaturated product detected by IRRAS is possible to be DOPC hydroperoxide."

P16L4–20 This material is more suitable for an introduction than a concluding section. Response:

We have moved this material to the introduction section and modified the atmospheric implication on Page3-4.

P16L21 I don't think that this statement regarding DOPC hydroperoxides is supported by the data presented in the paper, given that DOPC hydroperoxides aren't actually measured.
Response:
We have modified the statement on Page 18:

"According to the mechanism of photosensitizing reaction, the possible products——DOPC hydroperoxides were more water-soluble. They appear to dissolve into bulk artificial seawater and partition into the hydrophilic core of organic aqueous aerosols."

P17L11 DOPC hydroperoxide was not actually measured—in general, I think that this conclusions section does not accurately reflect the results presented in the paper.
Response:
We have modified the sentence in the conclusion on Page 18:

"The relative areas of DOPC monolayers in the artificial seawater containing photosensitizers were increased after irradiation. In addition, the largest increase of the relative area of the DOPC monolayer was observed in the presence of IC, as compared to the laboratory generated SOA sample and field collected PM$_{2.5}$ sample. The changes between the irradiated and dark PM-IRRAS spectra also support the photochemical oxidation of the unsaturated chains in DOPC."

---

## Author Response (AR2)

We have revised our manuscript according to the suggestions of the Referee's comments and the responses to the comments are as following. For clarity, the Referee's comments are reproduced in blue, authors' responses are in black and changes in the manuscript are in red color text.

Minor revisions:
Regarding the small spectral shifts presented as evidence for changes in DOPC conformation (e.g. P10L33, P11L7), the authors state in their response that "Spectra were averaged over 2000 scans, and IRRAS measurements were repeated at least three times to ensure reproducibility." What was the variability in the reported IRRAS peaks over the three trials? Was it smaller than the reported spectral shifts (i.e. < 2 cm-1)? I think that it would be very useful to clarify this in the text—I know that dark controls were performed, but the variability of the IRRAS bands for both the dark controls and the illuminated samples would be helpful to report.
Response:
We have taken irradiated and non-irradiated DOPC monolayer on the artificial seawater containing IC as examples and clarified the variability of the IRRAS bands for both the dark controls and the illuminated samples on Page 7. The figure of IRRAS variability have been added in the supplement:
"The variability in the IRRAS peaks over the three trials was smaller than 2 cm$^{-1}$, as shown in **Fig. S3**."

P1L1    "Photosensitizing compounds like brown carbon can absorb UV light"
BrC isn't a "compound"-this makes it sound like it is an individual chemical species, whereas it is actually a complex mixture of light-absorbing species.
Response:
We have modified the sentence on Page 1:
"Water-soluble brown carbon in the aqueous core of aerosol may play a role in the photochemical aging of organic film on the aerosol surface."

P1L1    "can absorb UV light and produce low volatile organic compounds (O:C ratio of 0.25 to 1)"
Where is this O:C range coming from? I wonder if a more general/"big picture" opening statement would be helpful here? Where are the low-volatility species coming from?
Response:
We have modified the sentence on Page 1:
"Water-soluble brown carbon in the aqueous core of aerosol may play a role in the photochemical aging of organic film on the aerosol surface."

P2L4
Literature citations for the fates of phospholipids would be helpful.
Response:
We have cited several references on Page 2 as suggested.

P3L5-20    "These water-soluble organic materials like IC are termed humic-like substances (HULIS) due to their similar properties to macromolecular humic substances"

I wouldn't classify IC itself as HULIS, since IC is a small-molecular-weight species, whereas HULIS are much higher in molecular weight.

Response:

We have modified the statement as suggested on Page 3.

"In addition to small-molecular-weight species like IC, water-soluble organic materials with higher molecular weight like humic-like substances (HULIS) can also absorb light. HULIS have the similar properties to macromolecular humic substances, such as their amphiphilic and polyacidic nature, aromaticity, surface active properties and light absorption ability (Gelencser et al., 2002; Graber and Rudich, 2006; Sannigrahi et al., 2006; Krivacsy et al., 2008)."

P5L7    "All size fractions of SOA we collected were involved in photosensitizing reaction."

Does this mean that all of the aluminum foil substrates were combined for extraction? This should be clarified in the text.

Response:

We have modified the sentence on Page 5-6:

"Then, all the aluminum foil substrates were combined for extraction."

P6L12-P7L6 "The collapse pressure of DOPC monolayer on pure artificial seawater decreased from 46 to 28 mN/m with the addition of IC. The collapse pressure for the DOPC monolayer on the artificial seawater containing HA was even lower."

It would be helpful here to explicitly state the concentrations, or at least mention that the collapse pressure would vary with concentration of the two additives—otherwise, the inference for the reader (I think) is that HA reduces the collapse pressure for DOPC more than IC.

Response:

We have modified the sentence on Page 9 as suggested.

"The collapse pressure of DOPC monolayer on pure artificial seawater decreased from 46 to 28 mN/m with the addition of 2.5 mM IC. The collapse pressure for the DOPC monolayer on the artificial seawater containing 30 mg/L HA was even lower."

"In contrast to IC and HA, the collapse pressures of DOPC monolayer were higher in the PM2.5 and SOA samples. The inorganic ions from the PM2.5 and SOA samples may contribute to the assembly of organic monolayer."

A reference for this statement would be helpful—how would the inorganic ions affect the assembly of the monolayer (what does assembly mean in this context)?

Response:

Inorganic ions can interact with lipids and produce ion-lipid complex through chelation (*Phys. Chem. Chem. Phys.*, **2016**, 18, 32345-32357; *Sci. Total Environ.*,

**2017**, 580, 1155-1161; *Phys. Chem. Chem. Phys.*, **2017**, 19, 10481-10490). The assembly of organic film here means the presence of inorganic ions in the $PM_{2.5}$ and SOA sample may induce the self-organization of lipid molecules and the stability of organic monolayer.

We have added some discussion and references on Page 9:

"In contrast to IC and HA, the collapse pressures of DOPC monolayer were higher in the $PM_{2.5}$ and SOA samples. The inorganic ions from the $PM_{2.5}$ and SOA samples like metal ions may contribute to the organization of organic monolayer (Adams et al., 2016; Adams et al., 2017; Denton et al., 2019)."